# The Underlying Universal Statistical Structure of Natural Datasets

**Noam Levi** [1]   **Yaron Oz** [2]

## Abstract

We study universal properties in real-world complex and synthetically generated datasets. Our approach is to analogize data to a physical system and employ tools from statistical physics and Random Matrix Theory (RMT) to reveal their underlying structure. Examining the local and global eigenvalue statistics of feature-feature covariance matrices, we find: *(i)* bulk eigenvalue power-law scaling vastly differs between uncorrelated Gaussian and real-world data, *(ii)* this power law behavior is reproducible using Gaussian data with long-range correlations, *(iii)* all dataset types exhibit chaotic RMT universality, *(iv)* RMT statistics emerge at smaller dataset sizes than typical training sets, correlating with power-law convergence, *(v)* Shannon entropy correlates with RMT structure and requires fewer samples in strongly correlated datasets. These results suggest natural image Gram matrices can be approximated by Wishart random matrices with simple covariance structure, enabling rigorous analysis of neural network behavior.

## 1. Introduction

Natural, or real-world, images are expected to follow some underlying distribution, which can be arbitrarily complex, and to which we have no direct access to. This distribution could have infinitely many nonzero moments, with varying relative importance compared to one another. In practice, we only have access to a very small subset of samples from the underlying distribution, which can be parameterized as $X \in \mathbb{R}^{d \times M}$, where $d$ is the dimension of each image vector and $M$ is the number of samples. The first moment of the data can always be set to 0, since we can remove the mean from each sample, without losing information

[1]École Polytechnique Fédérale de Lausanne (EPFL), Lausanne, Switzerland [2] Raymond and Beverly Sackler School of Physics and Astronomy, Tel-Aviv University, Tel-Aviv 69978, Israel. Correspondence to: Noam Levi <noam.levi@epfl.ch>.

*Proceedings of the 42$^{nd}$ International Conference on Machine Learning*, Vancouver, Canada. PMLR 267, 2025. Copyright 2025 by the author(s).

regarding the distribution. The second moment, however, cannot be set to 0, and holds valuable information. This observation motivates the study of the empirical covariance (Gram) matrix, $\Sigma_M = \frac{1}{M} X X^T$.

The properties of $\Sigma_M$ in real world data are entirely unknown a priori, as we do not know how to parameterize the process which generated natural images. Nevertheless, interesting observations have been made. Empirical evidence shows that the spectrum of $\Sigma_M$ for various datasets can be separated into a set of large eigenvalues ($\mathcal{O}(10)$), a bulk of eigenvalues which decay as a power law $\lambda_i \sim i^{-1-\alpha}$ (Ruderman, 1997; Caponnetto and De Vito, 2007) and a large tail of small eigenvalues which terminates at some finite index $n$. Since the top eigenvalues represent the largest overlapping properties across different samples, these are not simply interpreted without more information on the underlying distribution. The bulk of the eigenvalues, however, can be understood as representing the correlation structure of different features amongst themselves, and has been key to understanding the emergence of neural scaling laws (Kaplan et al., 2020; Maloney et al., 2022).

In this work, we study both the power law behaviors present in natural datasets, and their spectral statistics, with the goal of obtaining a universal, analytically tractable model for real world Gram matrices, regardless of their origins. While this may not be feasible for any $\Sigma_M$, fortunately, the standard datasets used today are high dimensional and contain many samples, a ubiquitous regime found in complex systems, and typically studied using Random Matrix Theory (RMT).

RMT is a powerful tool for describing the spectral statistics of complex systems. It is particularly useful for systems that are chaotic but also have certain coherent structures. The theory predicts universal statistical properties, provided that the underlying matrix ensemble is large enough to sufficiently fill the space of all matrices with a given symmetry, a property known as ergodicity (Guhr et al., 1998). Ergodicity has been observed in a variety of systems, including chaotic quantum systems (Bohigas et al., 1984; Mehta, 1991; Pandey, 1983), financial markets, nuclear physics and many others (Plerou et al., 1999; Brody, 1981; Efetov, 1997). To demonstrate that a similar universal structure is also observed for correlation matrices resulting from datasets, we will employ several diagnostic tools widely used in the field

of quantum chaos. We will analyze the global and local spectral statistics of empirical covariance matrices generated from three classes of datasets: (i) Data generated by sampling from a normal distribution with a specific correlation structure for its features, (ii) Uncorrelated Gaussian Data (UGD), (iii) Real-world datasets composed of images, at varying levels of complexity and resolution. Our research aims to answer the following questions:

- Is a power-law spectrum a universal property across real-world datasets?; what determines the scaling exponent and what properties should an analytical model of the dataset have in order to follow the same behavior?

- What universal properties of datasets can be gleaned from the empirical covariance matrix and how are they related to local and global statistical properties of RMT?

- How to quantify the extent to which complex data is well characterized by its Gram matrix?

- What, if any, are the relations between datasets power laws, entropy and statistical chaos diagnostics?

Our primary contributions are:

1. We find that power-law spectra appears across various datasets. It is governed by a single scaling exponent $\alpha$, and its origin is the strength of correlations in the underlying population matrix [1] . We accurately recover the behavior of the eigenvalue bulk of real-world datasets using Wishart matrices with the singular values of a Toeplitz matrix (Gray, 2006) as its covariance. We dub these *Correlated Gaussian Datasets* (CGDs).

2. We show that generically, the bulk of eigenvalues' distribution and spacings are well described by RMT predictions, verified by diagnostic tools typically used for quantum chaotic systems. This means that the CGD model is a correct proxy for real-world data covariances.

3. We find that the effective convergence of the empirical covariance matrix as a function of the number of samples correlates with the corresponding RMT description becoming a good description of the statistics and the eigenvalues power law decay.

4. The Shannon entropy is correlated with the local RMT structure and the eigenvalues behavior, and is substantially smaller in strongly correlated datasets compared to uncorrelated data. Additionally, it requires fewer samples to reach the distribution entropy.

---

[1]There are systems which display multiple correlation scales, showing several bulk exponents (Levi and Oz, 2023).

## 2. Background and Related Work

**Neural Scaling Laws**    Neural scaling laws are a set of empirical observations that describe the relationship between the size of a neural network, dataset, compute power, and its performance. These laws were first proposed by Kaplan et al. (2020) and have since been confirmed by a number of other studies (Maloney et al., 2022; Hernandez et al., 2022) and studied further in (Ivgi et al., 2022; Alabdulmohsin et al., 2022; Sharma and Kaplan, 2022; Sorscher et al., 2022; Debowski, 2023; Fernandes et al., 2023). The main finding of neural scaling laws is that the test loss of a neural network scales as a power-law with the number of parameters in the network. This means that doubling the number of parameters roughly reduces the test loss by $2^\alpha$. However, this relationship does not persist indefinitely, and there is a point of diminishing returns beyond which increasing the number of parameters does not lead to significant improvements in performance. One of the key challenges in understanding neural scaling laws is the complex nature of the networks themselves. The behavior of a neural network (NN) is governed by a large number of interacting parameters, making it difficult to identify the underlying mechanisms that give rise to the observed scaling behavior, and many advances have been made by appealing to the RMT framework.

**Random Matrix Theory**    RMT is a branch of mathematics that was originally developed to study the properties of large matrices with random entries. It is particularly suited to studying numerous realizations of the same system, where the number of realizations $M \to \infty$, the dimensions of the system $d \to \infty$, and the ratio between the two tends to a constant $d/M \to \gamma \leq 1, \gamma \in \mathbb{R}^+$. Results from RMT calculations have been applied to a wide range of problems in Machine Learning (ML), beyond the scope of neural scaling laws, including the study of nonlinear ridge regression (Pennington and Worah, 2017), random Fourier feature regression (Liao et al., 2021), the Hessian spectrum (Liao and Mahoney, 2021), and weight statistics (Martin and Mahoney, 2019; Thamm et al., 2022). For a review of some of the recent developments, we refer the reader to Couillet and Liao (2022) and references therein.

**Universality**    Considerable work has been dedicated to the concept of universality, i.e. that certain features are shared between seemingly disparate systems, when the systems are sufficiently large. For instance, spectra that are generated by different dynamical processes may have similar distributions (Bao et al., 2015; Baik et al., 2004; Hu and Lu, 2022; Bai and Silverstein, 2010). Universality is powerful since it often happens that System A's complex structure is difficult to analyze, and can be explained by system B, which lies in the same universality class, and is much easier to study. In our work, system A represents real-world datasets

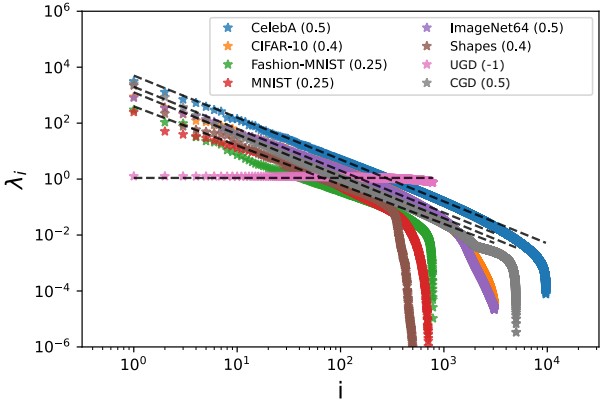
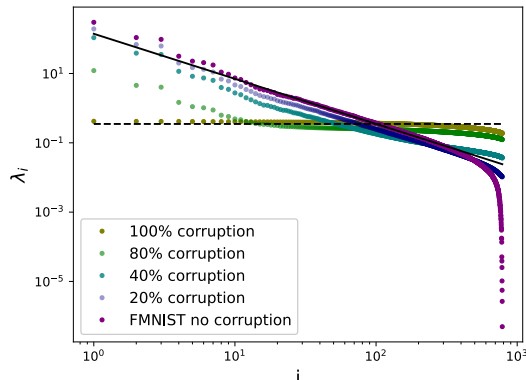

*Figure 1.* **Left:** Scree plot of $\Sigma_{ij,M}$ for several different vision datasets, as well as for UGD and a CGD with fixed $\alpha$. Here, the number of samples is taken to be the entire dataset for each real-world dataset, and $M = 50\text{k}$ for the gaussian data, where we set $c = 1$. We see a clear power law for the eigenvalue bulk as $\lambda_i \propto i^{-1-\alpha}$ where all real-world datasets display $\alpha \leq 1/2$. **Right:** The power-law exponent $\alpha$ value can be tuned from $\alpha = 1/4$ to $\alpha = -1$ by corrupting the FMNIST dataset with a varying amount of normally distributed noise.

with unknown statistics generated from a complex process, while system B is our CGD, whose Gram matrix is a simple Wishart matrix. The fact that real world datasets fall in the same universality class as CGD allows us to replace its complex covariance matrix by the simple CGD one, while retaining the information encoded in its spectrum.

## 3. Correlations and power-law spectra

In this section, we analyze the feature-feature covariance matrix for datasets of varying size, complexity, and origin. We consider real-world as well as correlated and uncorrelated gaussian datasets, establish a power-law decay of their eigenvalues, and relate it to a correlation length.

### 3.1. Feature-Feature Empirical Covariance Matrix

We consider the data matrix $\boldsymbol{X} = X_{ia} \in \mathbb{R}^{d \times M}$, constructed of $M$ columns, each corresponding to a single sample, composed of $d$ features. In this work, we focus on the empirical feature-feature covariance matrix, defined as

$$\Sigma_{ij,M} = \frac{1}{M} \sum_{a=1}^{M} X_{ia} X_{aj} \in \mathbb{R}^{d \times d} . \quad (1)$$

Intuitively, the correlations between the different input features, $X_{ia}$, should be the leading order characteristic of the dataset. For instance, if the $X_{ia}$ are pixels of an image, we may expect that different pixels will vary similarly across similar images. Conversely, the mean value of an input feature is uninformative, and so we will assume that our data is centered in a pre-processing stage.

A random matrix ensemble is a probability distribution on the set of $d \times d$ matrices that satisfy specific symmetry properties, such as invariance under rotations or unitary

transformations. In order to study Eq. (1) using the RMT approach, we define $\Sigma_{ij,a}$ as a single sample realization of the population random matrix ensemble $\Sigma_{ij}$, and thus $\Sigma_{ij,M}$ is the empirical ensemble average, i.e. $\Sigma_M = \langle \Sigma_a \rangle_{a \in M} = \frac{1}{M} \sum_{a=1}^{M} \Sigma_a$ approximating the limits of $M \to \infty, d \to \infty$. If $M$ and $d$ are sufficiently large, the statistical properties of $\Sigma_M$ will be determined entirely by the underlying symmetry of the ensemble. We refer to this case as the "RMT regime".

### 3.2. Data Exploration

We study the following real-world datasets: MNIST (LeCun et al., 2010), FMNIST (Xiao et al., 2017), CIFAR10 (cif), Tiny-IMAGENET (Torralba et al., 2008), and CelebA (Liu et al., 2015) (downsampeld to $109 \times 89$ in grayscale). We proceed to center and normalize all the datasets in the pre-processing stage, to remove the uninformative mean contribution. The uncorrelated gaussian data is represented by a data matrix whose elements $X_{ia} \in \mathbb{R}$, where each column is drawn from a jointly normal distribution $\mathcal{N}(0, \boldsymbol{I}_{d \times d})$. We then construct the empirical covariance matrix $\Sigma_M = \frac{1}{M} \sum_{a=1}^{M} X_{ia} X_{aj} \in \mathbb{R}^{d \times d}$. To generate correlated gaussian data, we repeat the same process, changing the sample distribution to $\mathcal{N}(0, \Sigma_{d \times d})$, where we choose a specific form for $\Sigma$ which produces feature-feature correlations and includes a natural cut-off scale, as

$$\Sigma_{ij}^{\text{Toe}} = S_{ij}, \quad T_{ij} = I_{ij} + c|i - j|^{\alpha} = (U^{\dagger}SV)_{ij}, \quad (2)$$

where $\alpha, c \in \mathbb{R}$. The matrix $\Sigma_{ij}^{\text{Toe}}$ is a positive semi definite diagonal matrix of singular values $S$ constructed from $T$, a full-band Toeplitz matrix. The sign of $\alpha$ dictates whether correlations decay (negative) or intensify (positive) with distance along a one-dimensional feature space[2].

---

[2]Correlation strength which grows with distance is a hallmark of some one-dimensional physical systems, such as the Coulomb

### 3.3. Correlations Determine the Spectral Noise to Data Transition

We begin by reproducing and extending some of the results from Maloney et al. (2022). In Fig. 1, we show the $\Sigma_{ij,M}$ eigenvalue power law decay for the different classes of data (*i.e.* real-world, UGD and CGDs). We find that for all datasets, the eigenvalues bulk scales as a power-law

$$\lambda_i \propto i^{-1-\alpha}, \quad \alpha \in \mathbb{R}, \quad i = 10, \ldots d_{\text{bulk}}, \qquad (3)$$

where $i = 10$ is approximately where the power law behavior begins and $d_{\text{bulk}}$ is the effective bulk size, where the power-law abruptly terminates. We stress that this behavior repeats across all datasets, regardless of origin and complexity.

The value of $\alpha$ can be readily explained in terms of correlations within our CGD model. Taking the Laplace Transform of the second term in Eq. (2), the bulk spectrum is given by Appendix B as

$$\lambda_i^{\text{bulk}} = c \cdot \Gamma(\alpha + 1) \left(\frac{d}{i}\right)^{1+\alpha}, \qquad (4)$$

where $\Gamma(x)$ is the Gamma function. This implies that the value of $\alpha$ determines the strength of correlations in the original data covariance matrix. For real-world data, we consistently find that $\alpha > 0$, which corresponds to increasing correlations between different features. In contrast, for UGD, the value of $\alpha \sim -1$, and the power-law behavior vanishes. Interpolating between UGD, and real-world-data, the CGD produces a power-law scaling, which can be tuned from $-1 < \alpha \leq 0$, in the case of decaying long range correlations, or $0 \leq \alpha < \infty$ for increasing correlations, to match any real-world dataset we examined. Lastly, we can extend this statement further and verify the transition from correlated to uncorrelated features by corrupting a real-world dataset (FMNIST) and observing the continuous deterioration of the power-law from $\alpha \sim 1/4$ to $\alpha = -1$, implying that the CGD can mimic the bulk behavior of both clean and corrupted data.

## 4. Global and Local Statistical Structure

### 4.1. Random Matrix Theory

In this section, we move on from the eigenvalue scaling behavior, to their statistical properties. We begin by describing the RMT diagnostic tools, often used to characterize RMT ensembles, with which we obtain our main results. We define the matrix ensemble under investigation, then provide an overview of each diagnostic, concluding with a summary

and Riesz gases (Lee and Yang, 1966; Smorodinsky, 1953), which display an inverse power-law repulsion, while decaying correlations are common in the 1-d Ising model (Ising, 1925).

of results for the specific matrix ensemble to which both real-world and Gaussian datasets converge.

We interpret $\Sigma_M$ for real world data as a single realization, drawn from the space of all possible Gram matrices which could be constructed from sampling the underlying population distribution. In that sense, $\Sigma_M$ itself is a random matrix with an unknown distribution. For such a random matrix, there are several universality classes, which depend on the strength of correlations in the underlying distribution. These range from extremely strong correlations, which over-constrain the system and lead to the so called Poisson ensemble (Atas et al., 2013), to the case of no correlations, which is equivalent to sampling independent elements from a normal distribution, represented by the Gaussian Orthogonal Ensemble (GOE) (Mehta, 2004). These classes are the only ones allowed by the symmetry of the matrix $XX^T$, provided that the number of samples and the number of features are both large. Since the onset of the RMT regime depends on the population statistics, it is a priori unknown. Determining if real data covariances converge to an RMT class, and to which one they converge to at finite sample size would inform the correct way to model real-world covariances.

Below we review the tools used in our analysis. While we provide an overview of each diagnostic, we refer the reader to Tao (2012); Kim et al. (2023) for a more comprehensivereview. We then apply these tools to gain insights into the statistical structure of the datasets.

**Spectral Density:** The empirical spectral density of a matrix $\Sigma$ is defined as,

$$\rho_\Sigma(\lambda) = \frac{1}{n} \sum_{i=1}^{n} \delta(\lambda - \lambda_i(\Sigma)), \qquad (5)$$

where $\delta$ is the Dirac delta function, and the $\lambda_i(\Sigma), i = 1, \ldots, n$, denote the $n$ eigenvalues of $\Sigma$, including multiplicity. The limiting spectral density is defined as the limit of Eq. (5) as $n \to \infty$.

**Level Spacing Distribution and $r$-statistics:** The level spacing distribution measures the probability density for two adjacent eigenvalues to be in the spectral distance $s$, in units of the mean level spacing $\Delta$. The procedure for normalizing all distances in terms of the local mean level spacing is often referred to as unfolding. We unfold the spectrum of the empirical covariance matrix $\Sigma_M(\rho)$ by standard methods (Kim et al., 2023), reviewed in Appendix A. Ultimately, the transformation $\lambda_i \to e_i = \tilde{\rho}(\lambda_i)$ is performed such that $e_i$ shows an approximately uniform distribution with unit mean level spacing. Once unfolded, the level spacing is given by $s_i = e_{i+1} - e_i$, and its probability density function $p(s)$ is measured.

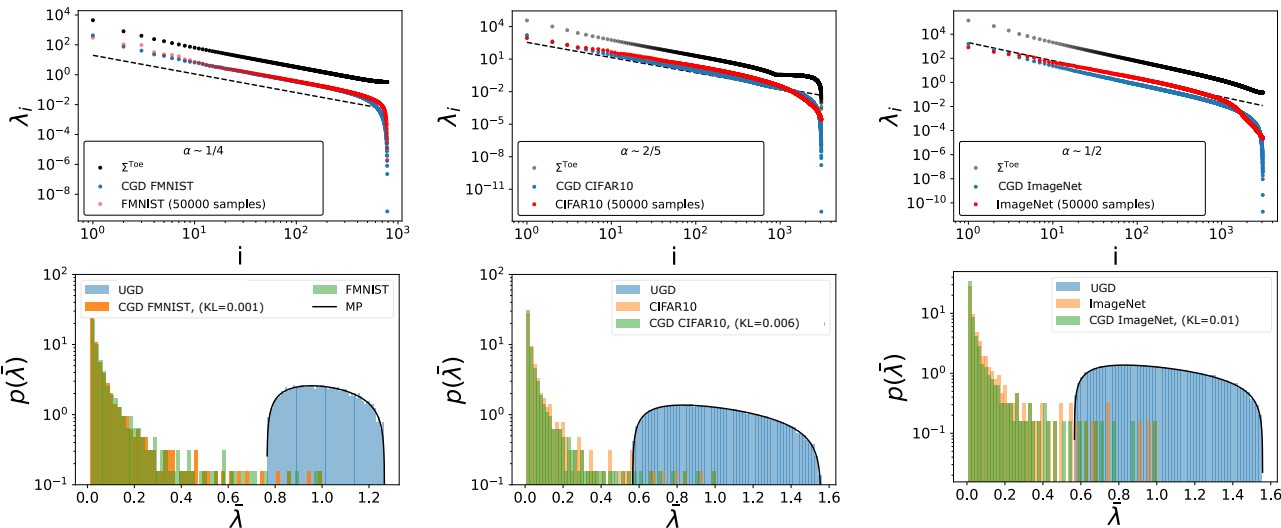

Figure 2. **Top row:** Scree plot of $\Sigma_{ij,M}$ for several different configurations and datasets. We show the eigenvalues of the population covariance matrix $\Sigma^{\text{Toe}}$, the eigenvalues for the empirical covariance of the full real-world dataset with $M = 50k$ and finally the eigenvalues of the empirical covariance using the same $\Sigma^{\text{Toe}}$, with $M = 50k$. The datasets used here are (left to right): FMNIST, CIFAR10, ImageNet. **Bottom row:** Spectral density for the bulk of eigenvalues for the same datasets, as well as a comparison against UGD of the same dimensions. The $\bar{\lambda}$ indicates normalization over the maximal eigenvalue among the bulk. We also provide the KL divergence between the CGDs and the real-world data distributions.

The distribution $p(s)$ captures information about the short-range spectral correlations, demonstrating the presence of level repulsion, *i.e.*, whether $p(s) \to 0$ as $s \to 0$, which is a common trait of the GOE ensemble, as the probability of two eigenvalues being exactly degenerate is zero. Furthermore, the level spacing distribution $p(s)$ for certain systems is known. For integrable systems, it follows the Poisson distribution $p(s) = e^{-s}$, while for chaotic systems (GOE), it is given by the Wigner surmise

$$p_\beta(s) = Z_\beta s^\beta e^{-b_\beta s^2}, \qquad (6)$$

where $\beta$, $Z_\beta$, and $b_\beta$ depend on which universality class of random matrices the covariance matrix belongs to (Mehta, 2004). In this work, we focus on matrices that fall under the universality class of the GOE, for which $\beta = 1$, as we show that both real-world data and CGD covariances belong to.

While the level spacing distribution depends on unfolding the eigenspectrum, which is only heuristically defined and has some arbitrariness, it is useful to have additional diagnostics of chaotic behavior that bypass the unfolding procedure. The $r$-statistics, first introduced in Oganesyan and Huse (2007), is such a diagnostic tool for short-range correlations, defined without the need to unfold the spectrum.

Given the level spacings $s_i$, defined as the differences between adjacent eigenvalues $\cdots < \lambda_i < \lambda_{i+1} < \cdots$ *without* unfolding, one defines the following ratios:

$$r_i = \text{Min}(s_i, s_{i+1})/\text{Max}(s_i, s_{i+1}), \quad 0 \le r_i \le 1 . \quad (7)$$

The expectation value of the ratios $r_i$ takes very specific values if the energy levels are the eigenvalues of random matrices: for matrices in the GOE the ratio is $\langle r \rangle \approx 0.536$. The value is smaller for integrable systems, approaching $\langle r \rangle \approx 0.386$ for a Poisson process (Atas et al., 2013).

**Spectral Form Factor:** The spectral form factor (SFF) is a long-range observable that probes the agreement of a given unfolded spectrum with RMT at energy scales much larger than the mean level spacing. It can be used to detect spectral rigidity, which is a signature of the RMT regime.

The SFF is defined as the Fourier transform of the spectral 2-point correlation function (Cotler et al., 2017; Liu, 2018)

$$K(\tau) = |Z(\tau)|^2/Z(0)^2 \simeq \frac{1}{Z}\langle |\sum_i \rho(e_i)e^{-i2\pi e_i \tau}\|^2 \rangle , \quad (8)$$

where $Z(\tau) = \text{Tr}e^{-i\tau\Sigma_M}$. The second equality is the numerically evaluated SFF (Juntajs et al., 2020), where $e_i$ is the unfolded spectrum, and $Z = \sum_i |\rho(e_i)|^2$ is chosen to ensure that $K(\tau \to \infty) \approx 1$.

The SFF has been computed analytically for the GOE ensemble, and it reads

$$K_{\text{GOE}}(\tau) = 2\tau - \tau\ln(1 + 2\tau) \text{ for } 0 < \tau < 1, \qquad (9)$$
$$K_{\text{GOE}}(\tau) = 2 - \tau\ln[(2\tau + 1)/(2\tau - 1)] \text{ for } 1 \le \tau .$$

Several universal features occur in chaotic RMT ensembles, manifesting in Eq. (9) and discussed in detail in Liu

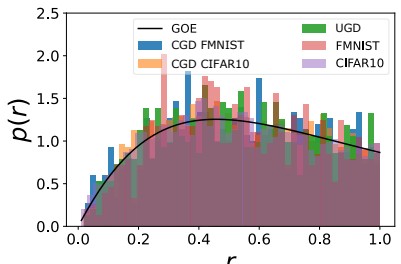 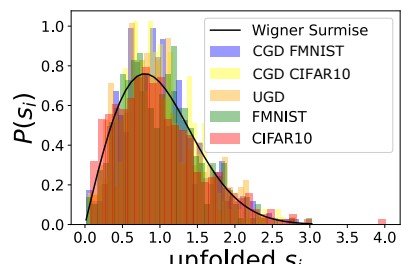 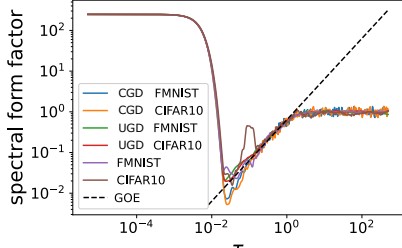

*Figure 3.* The $r$ probability density (**left**), the unfolded level spacing distribution (**center**) and the spectral form factor (**right**) of $\Sigma_M$ for FMNIST, CIFAR10, their CGDs, and UGD, obtained with $M = 50000$. Black curves indicate the RMT predictions for the GOE distribution from Eq. (11). These results indicate that the bulk of real-world data eigenvalues belongs to the GOE universality class, and that system has enough statistics to converge to the RMT predictions.

(2018); Kim et al. (2023). We mention here only two: (i) The constancy of $K(\tau)$ for $\tau \geq 1$ is simply a consequence of the discreteness of the spectrum. (ii) The existence of a timescale that characterizes the ergodicity of a dynamical system. It is defined as the time when the SFF of the dynamical system converges to the universal RMT computation. More concretely, it is indicated by the onset of the universal linear ramp $2\tau$ as in equation 9, which is absent in non-ergodic systems.

### 4.2. Insights from the Global and Local Statistics

#### 4.2.1. EIGENVALUES DISTRIBUTIONS

While the power law behavior of the bulk of eigenvalues is certainly meaningful, it is not the only piece of information that can be extracted from the empirical covariance matrix. Particularly, it is natural to inquire whether the origin of the power-law scaling determines also the degeneracy of each eigenvalue. We can test this hypothesis by comparing the global and local statistics of the bulk between real-world data and their CGD counterparts.

For the gaussian datasets we generate, there are known predictions for the spectral density, level spacing distribution, $r$-statistics and spectral form factor. In these special cases, the empirical covariance matrix in Eq. (1) is known as a Wishart matrix (Wishart, 1928): $\Sigma_{ij,M} \sim \mathcal{W}_d(\Sigma, M)$.

The spectral density $\rho(\lambda)$ of a Wishart matrix is given by the generalized Marčenko-Pastur (MP) law (Silverstein and Bai, 1995; Couillet and Liao, 2022), which depends on the details of $\Sigma$ and specified in App. B for certain limits. For $\Sigma = \sigma^2 I_d$, the spectral density is given explicitly by the MP distribution as

$$\rho(\lambda) = \frac{1}{2\pi\sigma^2} \frac{\sqrt{(\lambda_{\max} - \lambda)(\lambda - \lambda_{\min})}}{\gamma\lambda} , \quad (10)$$

for $\lambda \in [\lambda_{\min}, \lambda_{\max}]$ and 0 otherwise. Here, $\sigma \in \mathbb{R}^+$, $\lambda_{\max/\min} = \sigma^2(1 \pm \sqrt{\gamma})^2$, $\gamma \equiv d/M$ and $d, M \to \infty$.

In Fig. 11, we show that the CGDs capture not only the power law decay of the eigenvalue bulk, but also the spectral density and the distribution of eigenvalues, for ImageNet, CIFAR10, and FMNIST, measured by the Kullback–Leibler divergence (KL) (Kullback and Leibler, 1951). We further emphasize this point by contrasting the distributions with the MP distribution, which accurately captures the spectral density of the UGD datasets. This measurement alone is insufficient to determine that the system is well approximated by RMT, and we must study other statistical diagnostics.

#### 4.2.2. LEVEL SPACING DIAGNOSTICS

RMT predicts that certain local and global statistical properties are determined uniquely by symmetry. Therefore, the empirical covariance matrix must lie either in the GOE ensemble if it is akin to a quantum chaotic system[3] or in the Poisson ensemble, if it corresponds to an integrable system.

Both the level spacing and $r$ statistics (the ratio of adjacent level spacings) probability distribution functions and SFF for a Wishart matrix in the limit of $d, M \to \infty$ and $d/M = \gamma$, are given by the GOE universality class:

$$p_{\text{GOE}}(s) = \frac{\pi}{2}se^{-\frac{\pi}{4}s^2}, \quad \langle r \rangle_{\text{GOE}} = 4 - 2\sqrt{3}, \quad (11)$$
$$p_{\text{GOE}}(r) = \frac{27}{4}\frac{(r + r^2)}{(1 + r + r^2)^{5/2}}\Theta(1 - r),$$

In Fig. 3, we demonstrate that the bulk of eigenvalues for various real-world datasets behaves as the energy eigenvalues of a quantum chaotic system described by the GOE universality class. This result is matched by both the UGD and the CGDs, as is expected of a Wishart matrix. Here, the dataset size is taken to be $M = 50000$ samples, and the results show that this sample size is sufficient to provide a proper sampling of the underlying ensemble.

---

[3]Large random real symmetric matrices belong in the orthogonally invariant class.

### 4.2.3. HIGHER ORDER STATISTICS

Going beyond the nearest-neighbor local $r$ statistics, one can probe structure at increasingly larger scales. In particular, the $n$ neighbor local statistics can be defined as

$$r^{(n)} = \frac{\lambda_{i+2n} - \lambda_{i+n}}{\lambda_{i+n} - \lambda_i}, \tag{12}$$

for which the GOE distribution is known analytically (Tekur et al., 2018)

$$p_{\text{GOE}}(r^{(n)}) = Z \frac{(r + r^2)^\nu}{(1 + r + r^2)^{1+3\nu/2}}, \tag{13}$$

where $\nu = \frac{1}{2}n(n+3) - 1$ and $Z$ is a normalization factor. In Fig. 4, we show that the nearest-neighbor statistics are in good agreement with the GOE theory, but as the value of $n$ gets larger, farther neighbors are probed and discrepancies between the distributions emerge. Probing these higher order local statistics in greater detail can offer a systematic path towards explaining the discrepancies between the expected learning curves for neural network performance trained on gaussian vs. real world data (Loureiro et al., 2021), since $r^{(n)}$ statistics explicitly measure deviations from gaussianity.

### 4.2.4. EFFECTIVE CONVERGENCE

Having confirmed that CGDs provide a good proxy for the bulk structure for a large fixed dataset size, we may now ask how the statistical results depend on the number of samples.

As discussed in Sec. 3.1, $\Sigma_M$ can be interpreted as an ensemble average over single realizations of the true population covariance matrix $\Sigma$. As the number of realizations $M$ increases, a threshold value of $M_{\text{crit}}$ is expected to appear when the space of matrices that matches the effective dimension of the true population matrix is fully explored.

The specific value of $M_{\text{crit}}$ can be approximated without knowing the true effective dimension by considering two different evaluation metrics. Firstly, convergence of the local statistics of $\Sigma_M$, given by the point at which its level spacing distribution and $r$ value approximately match their respective RMT ensemble expectations. Secondly, convergence of the global spectral statistics, both of $\Sigma_M$ to that of $\Sigma$ and of the empirical parameter $\alpha_M$ to its population expectation $\alpha$.

Here, we define these metrics and measure them for different datasets, obtaining analytical expectations for the CGDs, which accurately mimic their real-world counterparts.

We can deduce $M_{\text{crit}}$ from the local statistics by measuring the difference between the empirical average $r$ value and the theoretical one given by

$$|r_M - r_{\text{RMT}}| = \delta(M) r_{\text{GOE}}, \tag{14}$$

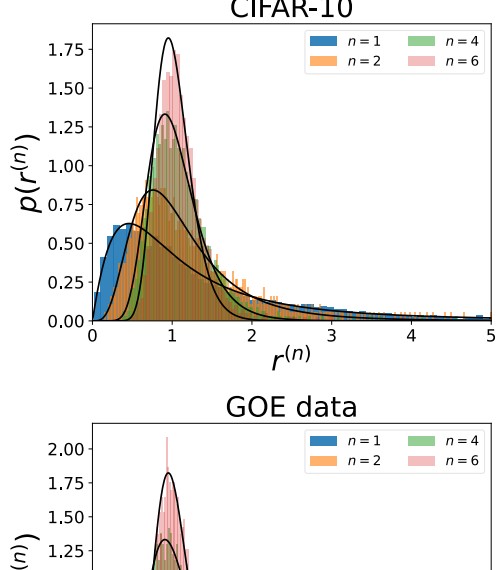

Figure 4. **Top:** The $r^{(n)}$ statistics for CIFAR10. **Bottom:** The $r^{(n)}$ statistics for GOE data. The local statistics at different scales $(n)$ are shown in different colors, while the *black* curves indicate the theoretical curves. For small $n = 1, 2, 4$ the $r^{(n)}$ distributions match between real-world data and the GOE results, but at larger scales $n = 6$ a deviation begins to appear.

where $r_{\text{GOE}} = 4 - 2\sqrt{3} \simeq 0.536$ for the GOE.

Next, we compare the results obtained for $M_{\text{crit}}$ from $\delta(M)$ to the ones obtained from the global statistics by using a spectral distance measure for the eigenvalue bulk given by

$$|\alpha_M - \alpha| = \Delta(M), \tag{15}$$

where $\alpha_M$ is the measured value obtained by fitting a power-law to the bulk of eigenvalues for a fixed dataset size $M$, while $\alpha$ represents the convergent value including all samples from a dataset.

Lastly, we compare the empirical Gram matrix $\Sigma_M$ with the convergent result $\Sigma$ obtained using the full dataset by taking

$$|\Sigma_M - \Sigma| = \epsilon(M)|\Sigma|, \tag{16}$$

where $|A|$ is the spectral norm of $A$, and $\epsilon(M)$ will be our measure of the distance between the two covariances.

In Fig. 6, we show the results for each of these metrics separately as a function of the number of samples $M$. We find that the $\delta(M)$ parameter, which is a measure of local

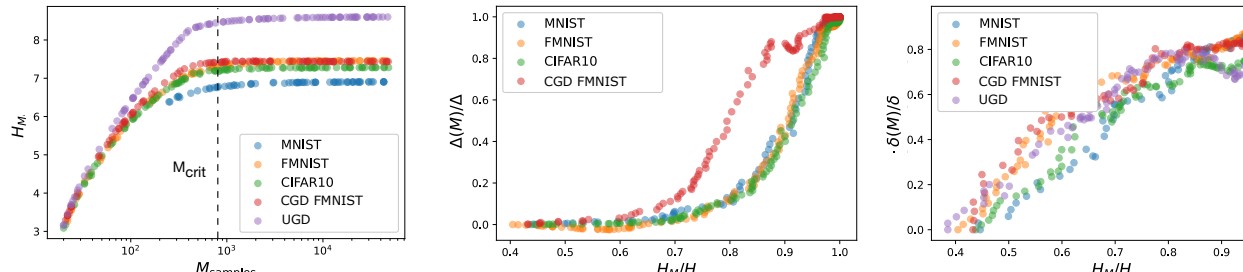

*Figure 5.* Convergence of the various metrics in Eqs. (14) to (16) in relation to entropy for the bulk of eigenvalues. **Left:** The Shannon entropy $H_M$ as a function of the dataset size $M$. **Center:** Convergence of the normalized $\alpha$ metric $\Delta_M/\Delta$ to its asymptotic value as a function of the normalized entropy $H_M/H$. **Right:** Convergence of the normalized $r$ statistics metric $\delta_M/\delta$ to its asymptotic value as a function of the normalized entropy $H_M/H$. We show the results for CIFAR10, FMNIST, MNIST, UGD, and the FMNIST CGD[5].

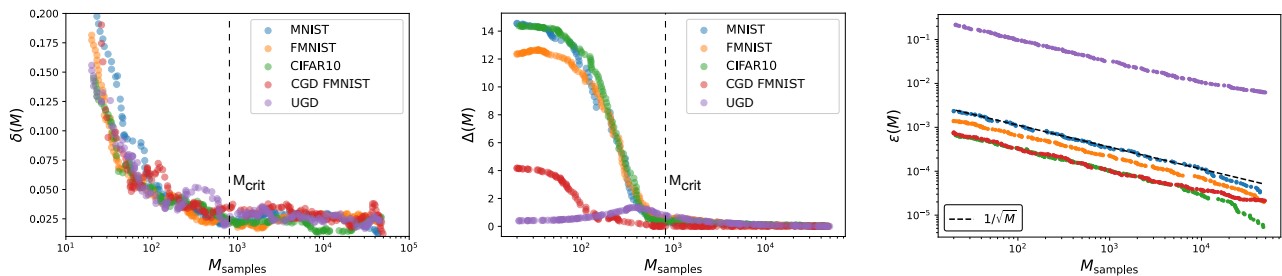

*Figure 6.* **Left:** The $r$ distance metric $\delta(M)$ for the bulk of eigenvalues. **Center:** The $\alpha$ distance metric $\Delta(M)$ for the bulk of eigenvalues. **Right:** The full matrix comparison metric $\epsilon(M)$. We show the results for CIFAR10, FMNIST, UGD, and the FMNIST CGD as a function of the number of samples. The results show that the bulk distances decrease as $1/M$, where $M$ is the number of samples, asymptoting to a constant value at similar values of $M_{\mathrm{crit}} \sim d$ (**black dashed**), where $d$ is the number of features.

statistics, converges to the expected GOE value at roughly the same $M_{\mathrm{crit}}$ as the entirely independent $\Delta(M)$ parameter, which measures the scaling exponent $\alpha$. The combination of these two metrics confirms empirically that the system has become ergodic at sample sizes roughly $M_{\mathrm{crit}} \sim d$, which is much smaller than the typical size of the datasets.

### 4.2.5. DATASETS ENTROPY

The Shannon entropy (Shannon, 1948) of a random variable a measure of information, inherent to the variable's possible outcomes (Rényi, 1956), given by $H = -\sum_{i=1}^{n} p_i \log(p_i)$ where $p_i$ is the probability of a given outcome and $n$ is the number of possible states. For covariance matrices, we define $p_i$ given the spectrum as $p_i = \lambda_i / \sum_{i=1}^{n_{\mathrm{bulk}}} \lambda_i$, where $n_{\mathrm{bulk}}$ is the number of bulk eigenvalues.

In Fig. 5 (left) we plot the Shannon entropies of real and gaussian datasets as a function of the number of samples. The entropies grow linearly and reach a plateau whose value is related to the correlation strength, with strong correlation corresponding to low entropy. We see the same entropy for both the gaussian and real datasets that have the same scaling exponent, implying that they also share the same eigenvalues degeneracy.

### 4.2.6. ENTROPY, SCALING EXPONENT AND RMT

In Fig. 5 (left), we see that the entropy saturation is correlated with the effective convergence in Fig. 6 as a function of the number of samples, while the middle and right plots show the correlation between the convergence of the entropy, the scaling exponent, and the r-statistics, respectively. We see that real data and gaussian data with the same scaling exponent exhibit similar convergence behavior.

### 4.3. Eigenvector Phenomenology

The results of our work concern the eigenvalue behavior of natural dataset covariances, as these seem to be well captured by a universal RMT description. The eigenvectors, however, specify the non-universal basis, which is highly dataset dependent, and should not be captured by a random orthogonal matrix. Nevertheless, for completeness, we explore the phenomenology of covariance eigenvectors in App. E. Our main observations are: (i) the eigenvectors obtained from an empirical covariance $\Sigma_M$ align, on average, with the population as $M^{1/2}$ for small sample sizes and then exponentially with $M$ (Fig. 12), and the transi-

---

[5]We omit UGD from the center panel, as $\alpha = -1$ for any $M$.

tion point $M_{\text{vec}}$ is significantly smaller than the eigenvalue threshold $M_{\text{crit}}$; (ii) the eigenvectors corresponding to the largest eigenvalues, as well as the smallest ones, align faster than the bulk of eigenvectors (Figs. 13 and 14), in contrast with the GCD model (Fig. 15), a property that seems to be related to how spatially localized these eigenvectors are. These results imply that accomplishing learning tasks for which the spectrum is insufficient and eigenvector properties are needed, requires a different scale of data samples.

## 5. Conclusions

In this work, we demonstrate that the bulk eigenvalue spectra of Gram matrices for real-world data can be accurately modeled by a Wishart matrix with a shift-invariant correlation structure and defining exponent $\alpha$. Crucially, these Gram matrices universally conform to the GOE statistics across various data-generating processes. This universality indicates that our approximation captures not only the scaling behavior of eigenvalues but also their full distribution, which can be rigorously derived using tools from RMT.

Our findings bridge a critical gap between theoretical models of data—often used to study neural network (NN) behavior—and real-world datasets. While Normally distributed data has advanced our understanding of NN learning dynamics (Gerace et al., 2021; Mei and Montanari, 2020; dAscoli et al., 2020), parameter scaling limits (Maloney et al., 2022), and weight evolution (Arous et al., 2018), such models inherently lack the structured correlations present in natural data. As a concrete application, we show in App. F that accurate modeling of spectral densities—enabled by our framework—is essential for solving the dynamics of even simple teacher-student models with correlated inputs.

We propose an RMT-based data model that preserves analytical tractability while incorporating realistic correlations. Although similar assumptions appear implicitly in neural scaling law studies, our work strengthens this paradigm with rigorous mathematical foundations, empowering practitioners to derive predictions better aligned with real data.

Our results also constrain the statistical properties of real-world data distributions. Specifically, the convergence of Gram matrices to GOE (rather than Poisson) statistics under finite sampling imposes non-trivial moment conditions on the underlying data distribution. This insight could inform both the inference of data-generating processes and the design of synthetic datasets that replicate natural correlations.

Although our analysis focuses on Gram matrices, which inherently discard spatial relationships, this simplicity enables a broad applicability. Extending our framework to language datasets, audio signals, or other modalities may reveal universal power-law behaviors transcending data types. Future work should explore the interplay between eigenvalues and

eigenvectors in neural networks, as both components critically shape information processing.

Finally, while we empirically observe chaotic spectral properties in real-world data, the origin of this chaos remains unclear. Is it intrinsic to the strongly correlated structure of natural data, or does it arise from noise in the sampling process? Resolving this question—potentially through controlled studies of synthetic datasets with tunable correlation and noise—will deepen our understanding of data complexity.

## Acknowledgements

We would like to thank Yohai Bar-Sinai, Marat Freytsis, Alex Maloney, Dan Roberts, and Jamie Sully for useful discussions and comments. N.L. would like to thank the Milner Foundation for the award of a Milner Fellowship. The work of N.L. is supported by the AI4Science program at EPFL. The work of Y.O. is supported in part by Israel Science Foundation Center of Excellence. This work was performed in part at Aspen Center for Physics, which is supported by the U.S. National Science Foundation grant PHY-2210452.

## Impact Statement

This paper presents work whose goal is to advance the field of Machine Learning. There are many potential societal consequences of our work, in particular due to the large model sizes considered in this work, but we do not feel there are specific aspects of this work with broader impacts beyond the considerations relevant to all large machine learning models.

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

## A. The Unfolding Procedure

Here, we provide additional details on the unfolding procedure used to produce Fig. 3 in the main text.

Care must be taken when analyzing the eigenvalues of the empirical covariance matrix $\Sigma_M$, since they exhibit unavoidable numerical errors. To control for the effect of numerical errors, we adopted a robust phenomenological procedure that utilizes the fact that all eigenvalues of $\Sigma_M$ must be non-vanishing by definition. To ensure we consider only eigenvalues of $\Sigma_M$ unimpacted by edge effects, we inspect only the bulk spectrum.

Restricting to the bulk removes many eigenvalues of $\Sigma_M$ as many are zero for small M. However, for larger M when $\Sigma_M$'s structure is clearly visible, this is not the case. The procedure ensures the eigenvalues kept are robust and not significantly impacted by numerical precision. From the significant eigenvalues of the empirical covariance matrix $\Sigma_M$, we compute the spectrum $\lambda_i$.

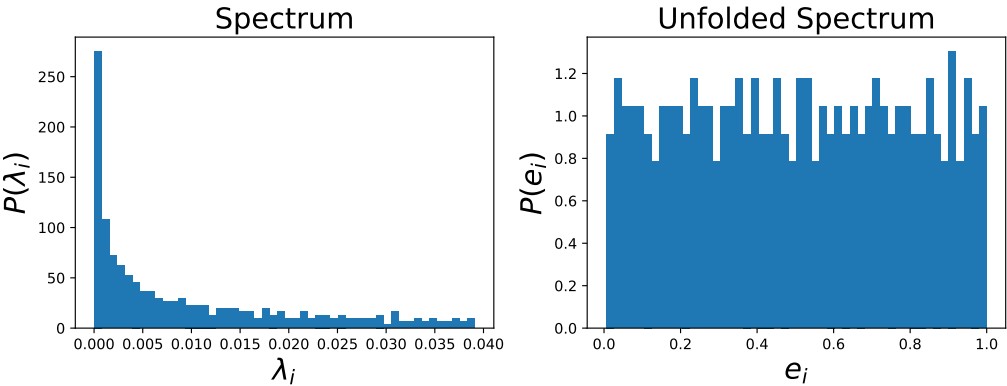

*Figure 7.* Bulk eigenvalue distribution for the empirical covariance matrix constructed from $M = 50000$ samples of FMNIST, before unfolding (**left**), and after unfolding (**right**) The unfolded spectrum displays approximately unit mean, and defined on the interval $[0, 1]$.

The unfolding procedure used to derive the unfolded spectrum is as follows:

1. Arrange the non-degenerate eigenvalues, $\lambda_i$ , of the empirical covariance matrix ( $\Sigma_M$) in ascending order.

2. Compute the staircase function $S(\lambda)$ that enumerates all eigenstates of the empirical covariance matrix ( $\Sigma_M$) whose eigenvalues are smaller than or equal to $\lambda$.

3. Fit a smooth curve, denoted by $\tilde{\rho}(\lambda)$ , to the staircase function. Specifically, we used a $12^{th}$-order polynomial as the smooth approximation.

4. Rescale the eigenvalues $\lambda_i$ as follows:
$$\lambda_i \to e_i = \tilde{\rho}(\lambda_i) \tag{17}$$

5. By construction, the unfolded eigenvalues $e_i$ should show an approximately uniform distribution with mean level spacing 1. This can be used to check if the procedure was successful by plotting the unfolded levels and checking the flatness of the distribution.

In Fig. 7, we show an example of the unfolding procedure for the FMNIST dataset. Specifically, we show the eigenvalue distribution before ($P(\lambda_i)$ ) and after ($P(e_i)$) unfolding. Up to the quality of the smoothing function $\tilde{\rho}(\lambda_i)$, the unfolded eigenvalue distribution displays a uniform distribution on the unit interval.

## B. Spectral Density for Wishart Matrices with a Correlated Features

For $z \in \mathbb{C}\backslash\text{supp}(\rho_\Sigma)$, the Stieltjes transform $G$ and inverse Stieljes transform $\rho_\Sigma$ are defined as

$$G(z) = \int \frac{\rho_\Sigma(t)}{z - t} dt = -\frac{1}{n}\mathbb{E}\big[\text{Tr}(\Sigma - zI_n)^{-1}\big], \quad \rho_\Sigma(\lambda) = -\frac{1}{\pi}\lim_{\epsilon \to 0^+} \Im G(\lambda + i\epsilon), \tag{18}$$

where $\mathbb{E}[\ldots]$ is taken with respect to the random variable $X$ and $(\Sigma - zI_n)^{-1}$ is the resolvent of $\Sigma$.

For the construction, discussed in the main text, and general $\alpha$, there is no closed form for the spectral density. However, in certain limits, analytical expressions can be derived from the Stieljes transform using Eq. (18). Specifically, given a determinstic expression for $\Sigma$, the spectral density can be derived by evoking Theorem 2.6 found in Couillet and Liao (2022), which uses the following result by Silverstein and Bai (1995)

$$G(z) = \frac{1}{\gamma}\tilde{G}(z) + \frac{1-\gamma}{\gamma z}, \quad \tilde{G}(z) = \left(-z + \frac{1}{M}\mathrm{Tr}\left[\Sigma(I_d + \tilde{G}(z)\Sigma)^{-1}\right]\right)^{-1}, \tag{19}$$

where $\gamma \equiv d/M$ and $d, M \to \infty$, and we substitute $C$ from the original theorem with $\Sigma$.

The empirical covariance matrix of the Gaussian correlated datasets discussed in the main text, is a Wishart matrix with a deterministic covariance, and thus fits the requirements of Theorem 2.6, where $\Sigma^{\mathrm{Toe}} = S$, $S = V^{\dagger}TU$, and $T_{i,j} = \boldsymbol{I}_{ij} + c|i-j|^{\alpha}$. In order to use Eq. (19), it is useful to first find the singular values of $T_{i,j}$. This can be done by using the discrete Laplace transform (extension of the Fourier transform), leading to

$$\Sigma^{\mathrm{Toe}}(s) = S(s) = 1 + c\mathrm{Li}_{-\alpha}\left(e^{-\frac{s}{d}}\right) - ce^{-s}\Phi\left(e^{-\frac{s}{d}}, -\alpha, d\right), \tag{20}$$

where $s = 1\ldots d$, $\Phi(x, k, a)$ is the Lerch transcendent, and $\mathrm{Li}(x)$ is the Poly-log function. Note that by the definition of $S$, Eq. (20) is a non-negative function of $s$. Because the identity matrix commutes with $\Sigma^{\mathrm{Toe}}$, we may substitute Eq. (20) in Eq. (19) to obtain

$$\tilde{G}(z) = \frac{1}{-z + \frac{\gamma}{d}S_d(\alpha)}, \tag{21}$$

where we define the sum $S_d(\alpha)$ to be

$$S_d(\alpha) = \sum_{s=1}^{d} \frac{\Sigma^{\mathrm{Toe}}(s)}{1 + \tilde{G}(z)\Sigma^{\mathrm{Toe}}(s)}. \tag{22}$$

Since the behavior of $\Sigma^{\mathrm{Toe}}(s)$ is intrinsically different for positive and very negative $\alpha$, we separate the two cases. First, consider the case of $\alpha < -1$, where correlations decay very quickly. In this scenario, the covariance matrix reduces to $\tilde{\Sigma}^{\mathrm{Toe}}(s) \simeq 1$.

Here, $S_d(\alpha)$ is given simply by the $\alpha \to \infty$ limit

$$S_d(\alpha \to -\infty) = \sum_{k=1}^{d} \frac{1}{1 + \tilde{G}(z)} = \frac{d}{1 + \tilde{G}(z)}, \tag{23}$$

which is precisely the case of $\Sigma = I_d$.

Solving Eq. (21) using the above result yields the following expression for $\tilde{G}(z)$

$$\tilde{G}(z) = \frac{-1 - z + \gamma - \sqrt{(-1-z+\gamma)^2 - 4z}}{2z}. \tag{24}$$

Finally, substituting Eq. (24) into Eq. (18) leads to the known Marčenko-Pastur (MP) law (Couillet and Liao, 2022)

$$\rho(\lambda) = \frac{1}{2\pi}\frac{\sqrt{(\lambda_{\max} - \lambda)(\lambda - \lambda_{\min})}}{\gamma\lambda} \quad \text{for } \lambda \in [\lambda_{\min}, \lambda_{\max}] \text{ and 0 otherwise}, \tag{25}$$

where $\lambda_{\max/\min} = (1 \pm \sqrt{\gamma})^2$.

The other interesting limit is that of $\alpha > -1$, in which the correlations do not decay quickly, and for $d \to \infty$ the Laplace transform of the Toeplitz matrix simplifies to

$$\Sigma^{\mathrm{Toe}}(s) \simeq c \cdot \Gamma(1+\alpha)\left(\frac{d}{s}\right)^{1+\alpha}. \tag{26}$$

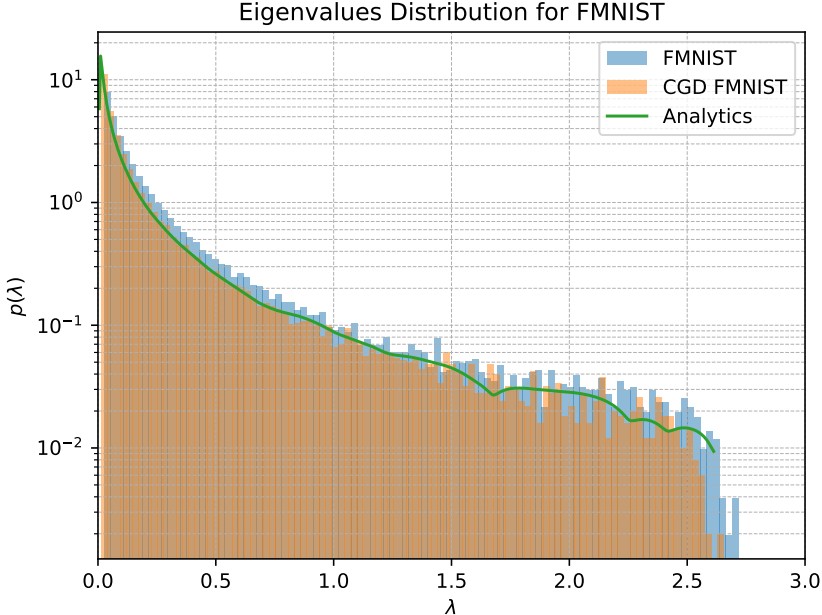

*Figure 8.* Theoretical predictions for the bulk spectral density of a CGD matrix against the empirical densities of FMNIST and the CGD. The **green** curve represents the generalized MP distribution, given by the solution to the inverse Steiljes transform in Eq. (29). The CGD curve has a value of $c = 1.14$, and $\gamma = 380/1000$ since that is the approximate number of bulk eigenvalues.

Using this approximation for the population covariance in Eq. (22) we obtain

$$\tilde{G}(z) = \left(-z + \frac{\gamma}{d}\sum_{s=1}^{d}\frac{c(s/d)^{-1-\alpha}}{1+\tilde{G}(z)c(s/d)^{-1-\alpha}}\right)^{-1}, \quad \hat{c} = c\Gamma(1+\alpha). \tag{27}$$

In the $d \to \infty$ limit, we can convert the sum to an integral using the Riemann definition

$$\lim_{d\to\infty}\frac{1}{d}\sum_{i=1}^{d}f(i/d) = \lim_{d\to\infty}\sum_{i=1}^{d}f(x_i)\Delta x = \int_{a}^{b}dxf(x), \quad \Delta x = \frac{b-a}{d} = \frac{1}{d}, \tag{28}$$

allowing us to write the equation for $\tilde{G}(z)$ as

$$\tilde{G}(z) = \left(-z + \frac{\gamma}{d}\sum_{s=1}^{d}\frac{\hat{c}(s/d)^{-1-\alpha}}{1+\tilde{G}(z)\hat{c}(s/d)^{-1-\alpha}}\right)^{-1} \simeq \left(-z + \gamma\int_{0}^{1}\frac{\hat{c}x^{-1-\alpha}}{1+\tilde{G}(z)\hat{c}x^{-1-\alpha}}dx\right)^{-1} \tag{29}$$

$$= \left(-z + \gamma\frac{{}_2F_1\left(1,\frac{1}{\alpha+1};1+\frac{1}{\alpha+1};-\frac{1}{\hat{c}\tilde{G}(z)}\right)}{\tilde{G}(z)}\right)^{-1},$$

where ${}_2F_1(a,b;c;z)$ is the Gaussian hypergeometric function. Eq. (29) is an algebraic equation which can be solved numerically, or analytically approximated in certain limits.

In Fig. 8, we show the theoretical results for the spectral density of a Wishart matrix with $\Sigma^{\mathrm{Toe}}$ covariance, for a value of $\alpha$ and $c$ matching FMNIST, against the empirical densities for FMNIST and the matching CGD. The green curve shows the generalized MP distribution given by the inverse Stieljes transform of Eq. (29).

## C. Robustness of the results

Here, we discuss some details regarding the robustness of our local and global statistical analyses.

For all of our analyses, we focused on the full Gram matrix, consisting of every sample in a given dataset. This implies that we only have access to a single realization of a $\Sigma_M$ empirical Gram matrix, per dataset, thus limiting our ability to perform standard statistics, for instance averaging over an ensemble of $\Sigma_M$, and obtaining confidence bands. This is not an issue in the RMT regime, as the matrix itself is thought of as an ensemble on to itself, and its eigenvalues have an interesting structure due to a generalization of the Central Limit Theorem (CLT).

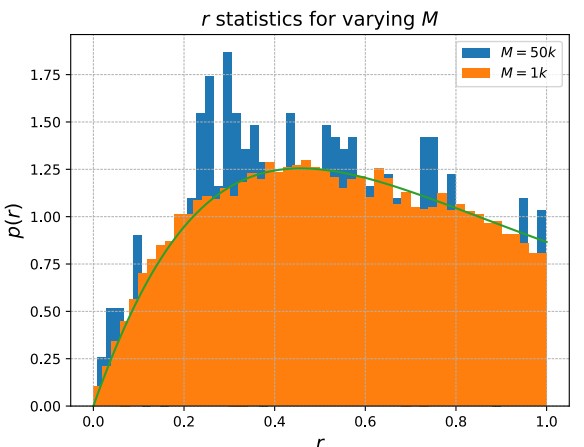
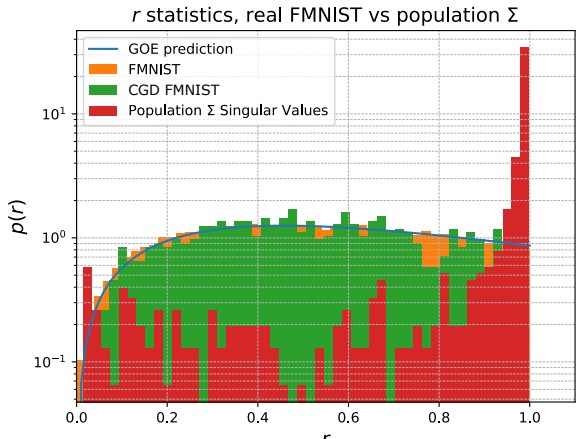

*Figure 9.* **Left:** The $r$ statistics distribution for FashionMNIST, comparing $M = 50000$ with $M = 1000$ subsets. In the first case, we obtain only a single realization of the Gram matrix, and so the $r$ statistics appear more noisy, however, when taking 40 realizations of a smaller subset, still above $M_{\text{crit}}$, we see that the fit to the GOE prediction (green) improves. We will add these figures, either in the main text or appendices, including goodness of fit measures on the rest of the datasets studied in the paper. **Right:** The $r$ statistics distribution for FashionMNIST and its CGD. In red, we show the singular values of the population covariance, $\Sigma^{\text{Toe}}$ used in the main text. In Orange, the true FMNIST $r$ distribution, obtained by taking 40 different realizations of a $M = 1000$ subset of the full dataset, leading to a perfect fit to the GOE prediction (blue). In green, we show the CGD using a 1000 samples as well. This figure illustrates that the deterministic population covariance does not sufficiently capture all the information that resides in the Gram matrix, while the CGD does.

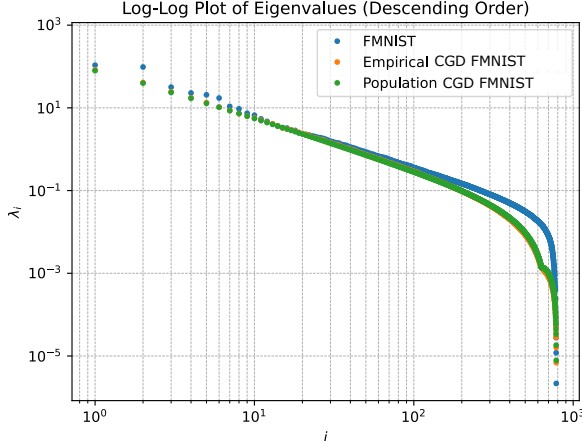
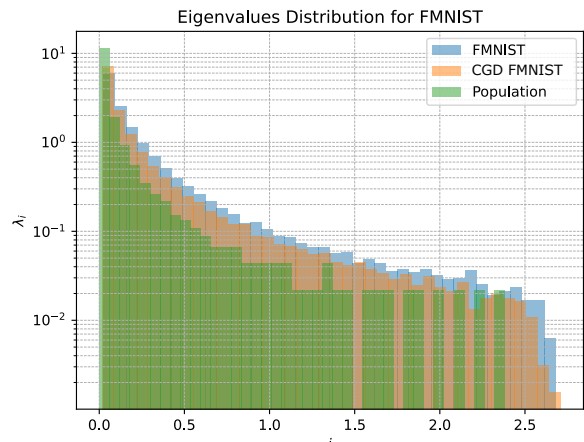

*Figure 10.* **Left:** Scree plot for the eigenvalues of the FashionMNIST Gram matrix (blue), its CGD (orange) using $M = 1000$ for 50 runs, and the Toeplitz population covariance matrix (green). Here, we show that the population and empirical covariance matrices match precisely in spectral scaling. The Gram matrices for FMNIST and its analogue are obtained by first normalizing the samples (mean subtraction and dividing by the standard deviation) and the population covariance is rescaled by a constant factor that depends only on the input dimension $d$. **Right:** The eigenvalue distribution for FashionMNIST, Gram matrix (blue), its CGD (orange) using $M = 1000$ for 50 runs, and the Toeplitz population covariance matrix (green). We see that the three distributions are similar, as can be expected, but that certain local features (such as the spacing between eigenvalues) is poorly captured by the deterministic population covariance.

We can still attempt to persuade the reader that our results are robust a posteriori, by noting that the number of samples

required to reach the RMT regime is approximately $M_{\text{crit}} \sim d$. This implies that we can consider sub-samples of the full empirical Gram matrix, consisting of $M_{\text{crit}}$, as $\Sigma_{M_{\text{crit}}} = 1/M_{\text{crit}} \sum_{a=1}^{M_{\text{crit}}} X_{ia} X_{aj}$, and average over multiple sub-sample matrices.

In Fig. 9, we see an implementation of this process for FMNIST, demonstrating that additional sampling pushes the distribution to a perfect fit for the GOE $r$-statistics, while in Fig. 10, we see the same type of convergence for the eigenvalue distribution.

## D. Results on Augmented Data

In Fig. 11, we show the RMT results for the eigenvalue spectra as well as the r-statistics for augmented versions of CIFAR10 and FashionMNIST. The augmentations include shifts, rescalings and rotations, in this case taken with a range of $50\%$ for height and width shift, rotation angle of $40$ degrees and zooming range up to $20\%$. The results indicate that global statisitics can change substantially, as seen in the the left scree plot of Fig. 11, while the local r-statistics is maintained even for highly augmented images.

## E. Eigenvector Phenomenology

In this section, we discuss the universal eigenvector behavior of two natural datasets: CIFAR10 and FashionMNIST. We compute the population (full training set size) and the empirical covariances $\Sigma, \Sigma_M$, and define their eigen-decomposition simpye as

$$\Sigma_M = U^T S U, \tag{30}$$

where $U$ is the matrix of eigenvectors and $S$ is a diagonal matrix composed of the eigenvalues of $\Sigma_M$. We further employ the magnitude of the cosine similarity to measure how aligned two eigenvectors are with one another, as

$$|\text{Cosine similarity}| = S_{C,ij} = \vec{u}_i \cdot \vec{u}_j, \tag{31}$$

which ranges between $[-1, 1]$, as the eigenvectors are normalized.

Analyzing the eigenvector alignment with the number of samples, we observe two interesting phenomena that seem to be universal. First, in Fig. 12 we show that for both datasets, the average magnitude $S_C$ behaves differently for small sample sizes, below and above some threshold value $M_{vec}$. Concretely, for $M < M_{vec}$ the similarity scales as $S_C \propto M^{1/2}$, while for $M > M_{vec}$, it scales as $S_C \propto e^{M/M_{pop}}$, where $M_{pop}$ is the population sample size, which is $M_{pop} = 50k$ for both datasets.

Secondly, in Fig. 13, we see a clear distinction between the eigenvector convergence behavior for a random basis (bottom rows) and natural data. The GCD model is always given in a random basis, whose eigenvectors align to the population eigenvectors in descending order, from the ones associated with the largest eigenvalues, to the ones associated with the smallest eigenvalues. A different phenomenon occurs for natural data - both the top eigenvectors and the smallest eigenvectors align together, while the bulk is aligned last. We believe this is tightly related to the localization properties of the eigenvectors. In Figs. 14 and 15 we see that the cosine similarity changes can be qualitatively traced to the the Inverse Participation Ratio (IPR), defined as

$$\text{IPR}_i = \sum_a \|u_{ia}\|^4, \tag{32}$$

where $i$ indicates the index of the eigenvector, while $a$ denotes the different elements of that particular eigenvector. A large small IPR implies that the vector is highly de-localized, since each element should scale as $1/\sqrt{d}$ and since there are $d$ elements, IPR $\sim 1/d$, while for a fully localized vector, all elements apart from one will be 0, therefore we have that the IPR $\sim 1$. We see in Figs. 14 and 15 that for natural data the IPR is small for the top eigenvectors, then plateaus before growing for the smallest eigenvectors, implying a transition from de-localization to localization. For CGD data, which has a random basis, the exact opposite occurs, where the first top eigenvectors are highly localized, and the rest are not.

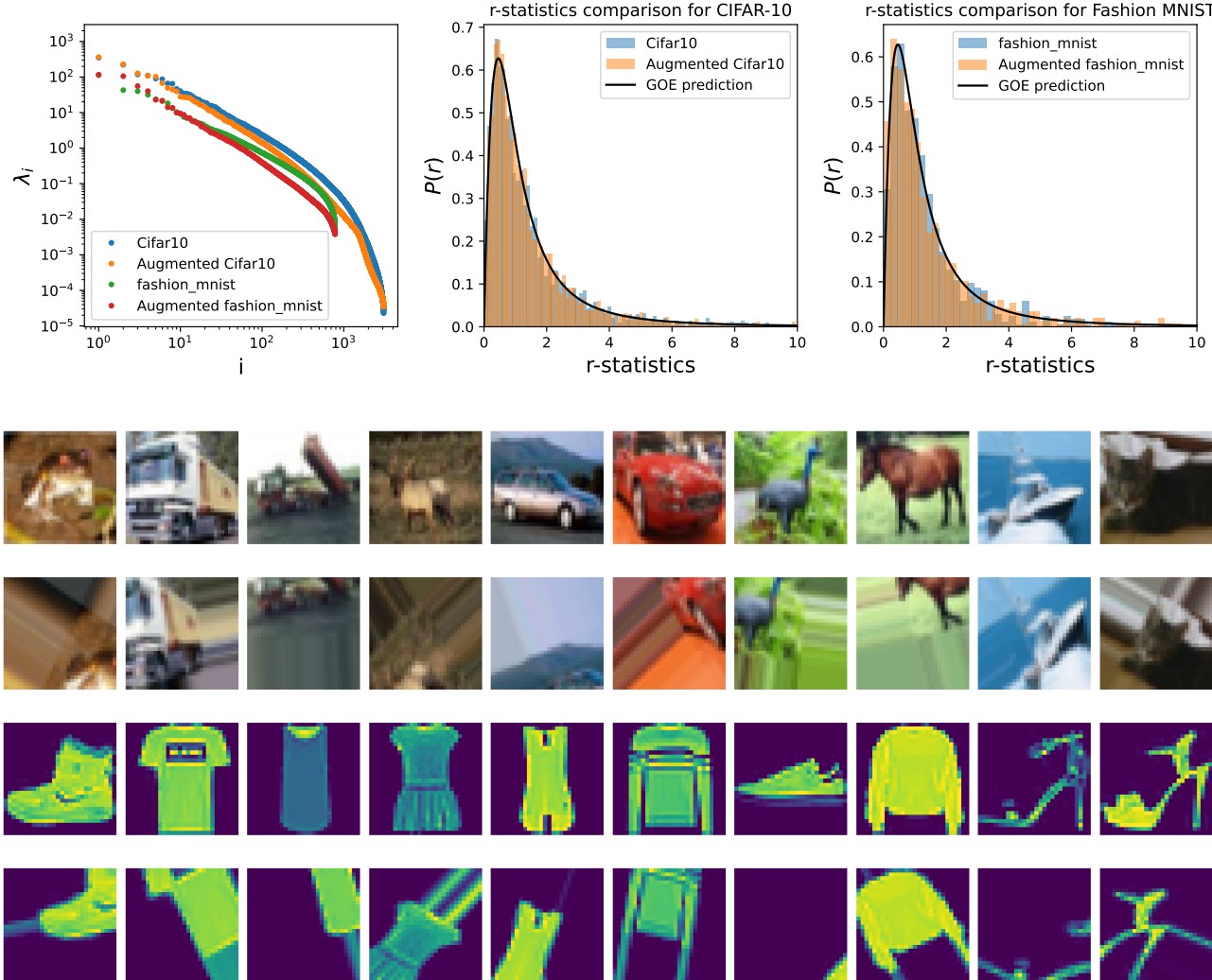

*Figure 11.* **Top:** Scree plot and $r$ statistics distributions for CIFAR10 and FashionMNIST, as well as augmented variations of these datasets for $M = 50k$ samples out of the datasets. Augmentations encompass shifts, rescalings and rotations, in this case taken with a range of $50\%$ for height and width shift, rotation angle of 40 degrees and zooming range up to $20\%$. Here, we use the definition of $r_i = s_i/s_{i-1}$, which has a known GOE prediction given by $P(r) = \frac{27}{8}\frac{(r+r^2)}{(1+r+r^2)^{5/2}}$. This can be easily translated to the form presented in the main text. **Bottom:** Representative images taken from the 4 types of datasets, in order from top row to bottom row are: CIFAR10, Augmented CIFAR10, FashionMNIST and Augmented FashionMNIST. The results show that the local statistical structure is maintained for the augmented parameters chosen, as well as a power law behavior for the eigenvalue bulk, but with different scaling exponent between the augmented and original data. It should be clear that if the parameters are exaggerated, for instance if the zoom range grows to the level of a constant gradient across the image, we do not expect the statistical structure to remain the same.

## F. Universality in Neural Network Analysis - A Toy Example

Universality laws have been employed in various ways to study error universality in neural networks, for instance in (Mei and Montanari, 2022; Gerace et al., 2020; Goldt et al., 2022; Hu and Lu, 2022). In this context, one looks directly at the universality of the training and generalisation errors instead of the features, taking into account the labels and the task. It has also been observed to hold for correlated Gaussian data in teacher-student settings in (Loureiro et al., 2021). Furthermore, it has been shown that for a simple regression task, the computation of the error reduces to an RMT problem (Wei et al., 2022), which is linked to the work presented in the main text regarding the data features themselves. In particular, it has been noted that in some cases the structure of the bulk fully characterises the error, even for multi-modal distributions, see (Gerace

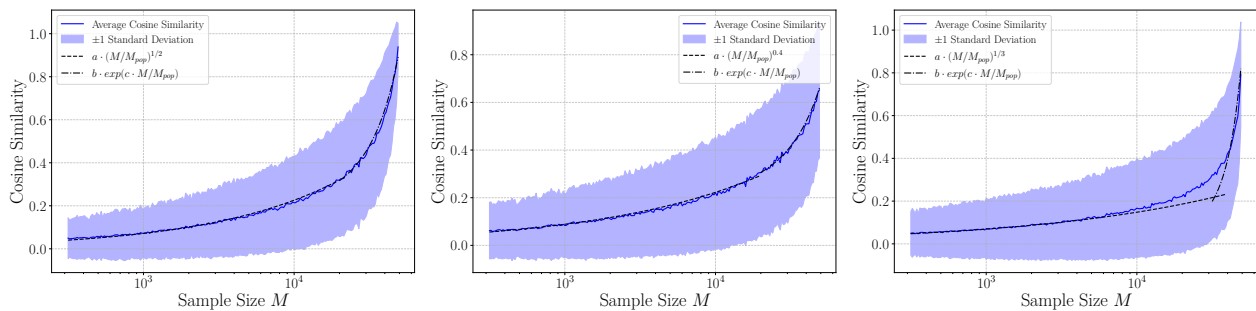

*Figure 12.* Average magnitude of the cosine similarity between population eigenvectors and their empirical estimate from $M$ samples. **Left:** The results for CIFAR10 eigenvectors. **Middle:** The results for FashionMNIST eigenvectors. **Right:** The results for CGD ($\alpha = 0.3$) eigenvectors. The results quite universally show an increase in cosine similarity that scales roughly as a power law ($M^{1/2}$ for real data and $M^{1/3}$ for CGD) for $M$ below some threshold value $M_{vec}$, while the increase from this value and above grows exponentially.

et al., 2023; Pesce et al., 2023).

As a self contained example of applying universality in neural network analysis, we call upon an exceedingly simple machine learning setup, namely, training a linear network using gradient descent to learn a teacher-student mapping. We show that even in this basic example, it is necessary to apply the results demonstrated in the main text in order to correctly analyze the system, when data correlations are present in the underlying population covariance.

Teacher-student models have been the subject of a long line of works (Seung et al., 1992; Watkin et al., 1993; Engel and Van den Broeck, 2001; Donoho, 1995; El Karoui et al., 2013; Saxe et al., 2014; Zdeborov'a and Krzakala, 2016; Donoho and Montanari, 2016) , and have experienced a resurgence of interest in recent years (Mei and Montanari, 2019; Hastie et al., 2019; Cand'es et al., 2020; Aubin et al., 2020; Salehi et al., 2020) as a powerful tool to study the high-dimensional asymptotic performance of learning problems with synthetic data.

The teacher-student model can be described as follows: The teacher uses ground truth information along with a probabilistic model to generate data which is then passed to the student who is supposed to recover the ground truth as well as possible only from the knowledge of the data and the model.

Here, we consider a linear teacher-student model, where the data inputs $x_i \in \mathbb{R}^{d_{\text{in}}}$ are identical independently distributed (iid) normal variables drawn from a Gaussian distribution with non-trivial population covariance $x_i \sim \mathcal{N}(0, \Sigma_{\text{pop}})$. We draw $N_{\text{tr}}$ training samples, and the teacher model generates output labels by computing a vector product on each input $y = w^* \cdot x$, where $w* \in \mathbb{R}^{d_{\text{in}}}$, assuming a perfect, noiseless teacher. The student, which shares the same model as the teacher, generates predictions $\hat{y} = w \cdot x$, where $w \in \mathbb{R}^{d_{\text{in}}}$ as well. The loss function which measures convergence of the student to the teacher outputs is the standard MSE loss. Our analysis is done in the regime of large input dimension and large sample size, i.e., $d_{\text{in}}, N_{\text{tr}} \to \infty$, where the ratio $\lambda \equiv d_{\text{in}}/N_{\text{tr}} \in \mathbb{R}^+$ kept constant. The student model is trained with the full batch Gradient Descent (GD) optimizer for $t$ steps with a learning rate $\eta$. The training loss function is given by

$$\mathcal{L}_{\text{tr}} = \frac{1}{N_{\text{tr}}} \sum_{i=1}^{N_{\text{tr}}} \|(w - w^*)^T x_i\|^2 = \text{Tr}\left[\Delta^T \Sigma_{\text{tr}} \Delta\right], \tag{33}$$

where we define $\Delta \equiv w - w^*$ as the difference between the student and teacher vectors. Here, $\Sigma_{\text{tr}} \equiv \frac{1}{N_{\text{tr}}} \sum_{i=1}^{N_{\text{tr}}} x_i x_i^T$ is the $d_{\text{in}} \times d_{\text{in}}$ empirical data covariance, or Gram matrix for the *training* set. The elements of $w^*$ and $w$ are drawn at initialization from a normal distribution $w_0, w^* \sim \mathcal{N}(0, 1/(2d_{\text{in}}))$. We do not include biases in the student or teacher weights, as they have no effect on centrally distributed data.

The generalization loss function is defined as its expectation value over the input distribution, which can be approximated by the empirical average over $N_{\text{gen}}$ randomly sampled points

$$\mathcal{L}_{\text{gen}} = \mathbb{E}_{x \sim \mathcal{N}}\left[\|(w - w^*)^T x\|^2\right] = \text{Tr}\left[\Delta^T \Sigma_{\text{gen}} \Delta\right]. \tag{34}$$

Here $\Sigma_{\text{gen}}$ is the covariance of the generalization distribution. Note that in practice the generalization loss is computed by a sample average over an independent set, which is not equal to the analytical expectation value. The gradient descent

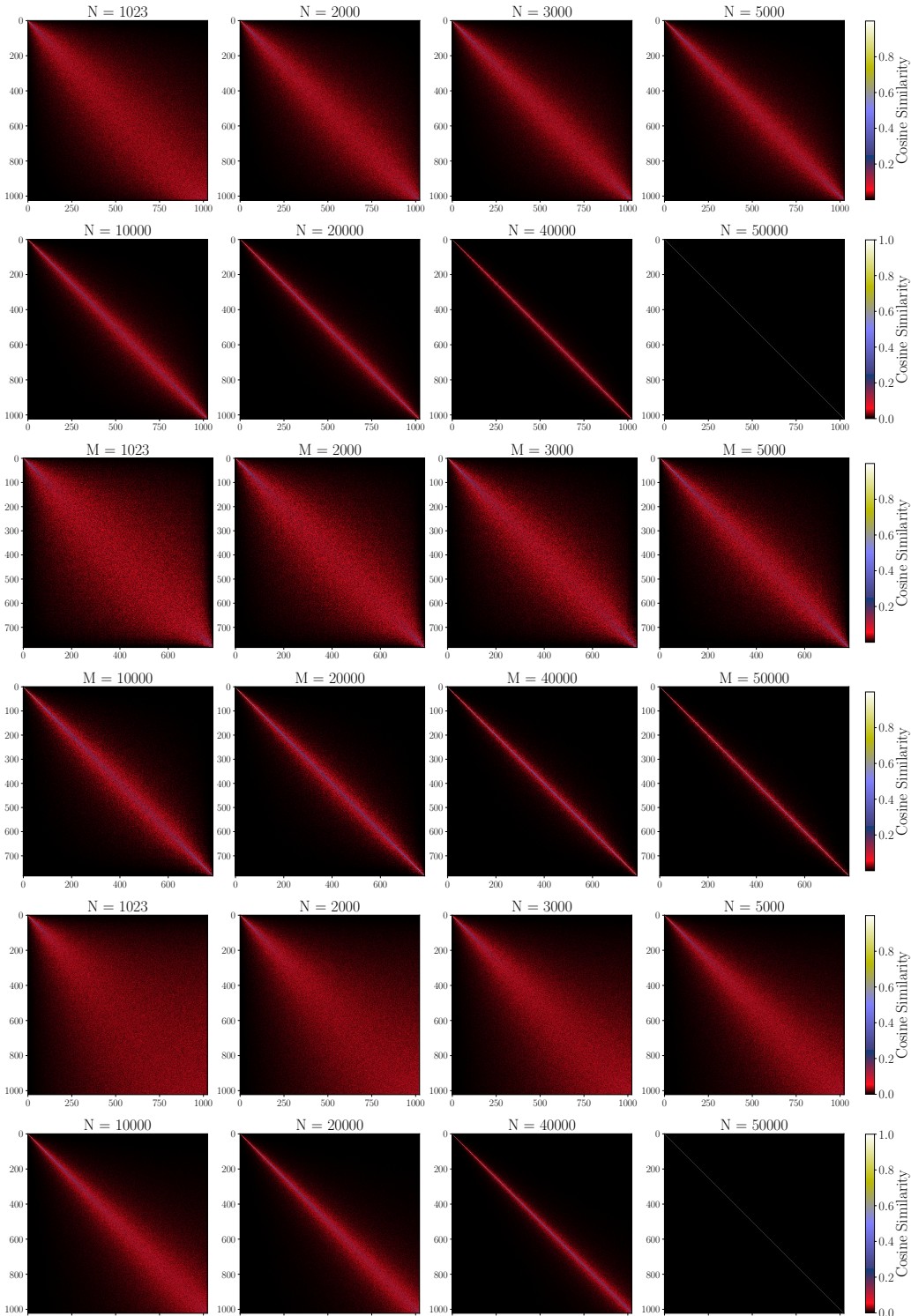

*Figure 13.* **Top 2 rows:** Covariance eigenvector cosine similarity magnitude matrices as a function of the number of samples $M$, for CIFAR10. **Middle 2 rows:** Covariance eigenvector cosine similarity matrices as a function of the number of samples $M$, for FashionMNIST. **Bottom 2 rows:** Covariance eigenvector cosine similarity matrices as a function of the number of samples $M$, for GCD with $\alpha = 0.3$. Clearly, a random basis aligns by first matching the eigenvectors associated with the largest eigenvalues, while natural data aligns first both the largest and the smallest eigenvectors.

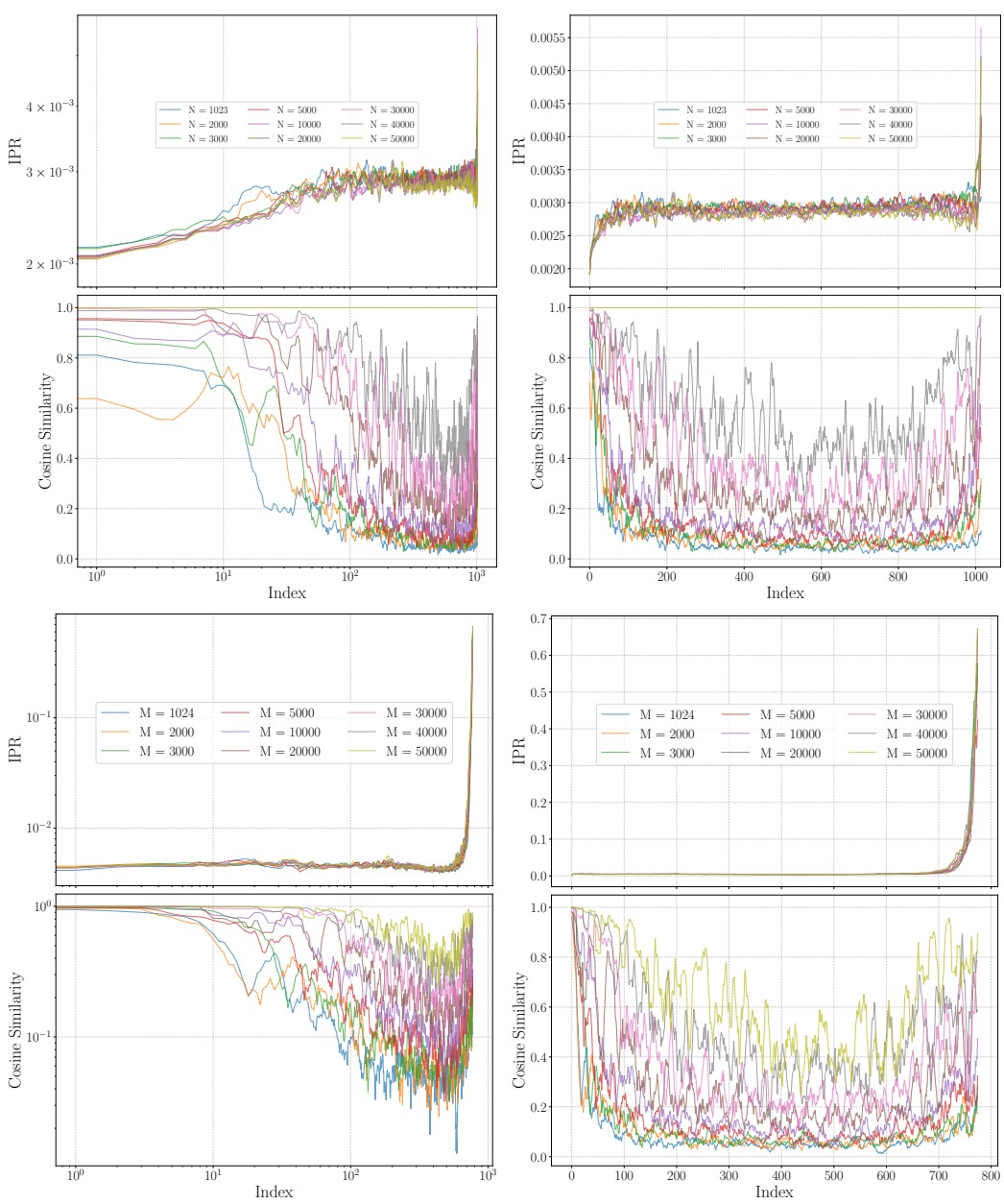

*Figure 14.* Cosine similarity for between each population eigenvector and its empirical estimate from $N$ samples, as well as IPR for each eigenvector as a function of number of samples. **Top row:** IPR in log and linear scale for CIFAR10. **Second row:** Absolute cosine similarity for CIFAR10. **Third row:** IPR in log and linear scale for FashionMNISt. **Fourth row:** Absolute cosine similarity for FashionMNIST.

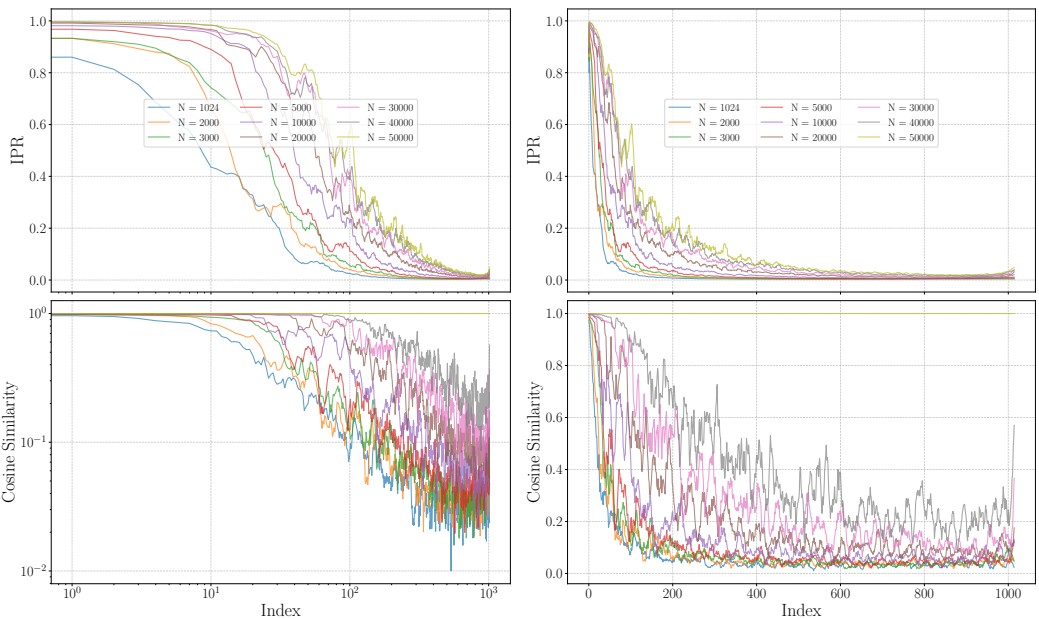

*Figure 15.* Cosine similarity for between each population eigenvector and its empirical estimate from $N$ samples, as well as IPR for each eigenvector as a function of number of samples. **Top row:** IPR in log and linear scale for CGD with $\alpha = 0.3$. **Bottom row:** Absolute cosine similarity in log and linear scale for CGD with $\alpha = 0.3$.

equations at training step $t$ are

$$\Delta_{t+1} = \left(\boldsymbol{I} - 2\eta\Sigma_{\text{tr}}\right)\Delta_t, \tag{35}$$

where $\gamma \in \mathbb{R}^+$ is the weight decay parameter, and $\boldsymbol{I} \in \mathbb{R}^{d_{\text{in}} \times d_{\text{in}}}$ is the identity.

Eq. (35) can be solved in the gradient flow limit, setting $\eta = \eta_0 dt$ and $dt \to 0$, resulting in

$$\dot{\Delta}(t) = -2\eta_0\Sigma_{\text{tr}}\Delta(t) \quad \rightarrow \quad \Delta(t) = e^{-2\eta_0\Sigma_{\text{tr}}t}\Delta_0, \tag{36}$$

where $\Delta_0$ is simply the difference between teacher and student vectors at initialization. It follows that the empirical losses, calculated over a dataset admit closed form expressions as

$$\mathcal{L}_{\text{tr}} = \Delta_0^T e^{-4\eta_0\Sigma_{\text{tr}}t}\Sigma_{\text{tr}}\Delta_0, \quad \mathcal{L}_{\text{gen}} = \Delta_0^T e^{-2\eta_0\Sigma_{\text{tr}}t}\Sigma_{\text{pop}}e^{-2\eta_0\Sigma_{\text{tr}}t}\Delta_0. \tag{37}$$

Since the directions of both $\Delta$ and the eigenvectors of $\Sigma_{\text{tr}}$ are uniformly distributed, we make the approximation that the projection of $\Delta$ on all eigenvectors is the same, which transforms Eq. (37) to the simple form

$$\mathcal{L}_{\text{tr}} \approx \frac{1}{d_{\text{in}}} \sum_i e^{-4\eta_0\nu_i t}\nu_i , \tag{38}$$

while the calculation for the generalization loss amounts to

$$\mathcal{L}_{\text{gen}} \approx \frac{1}{d_{\text{in}}} \sum_{i,j} e^{-2\eta_0(\nu_i+\nu_j)t}(U\Sigma_{\text{pop}}U^\dagger)_{ij} , \tag{39}$$

where $U$ is a random unitary matrix used to rotate to the basis of $\Sigma_{\text{tr}}$.

Now we turn to the choice of $\Sigma_{\text{pop}}$ and the implication for $\Sigma_{\text{tr}}$. As demonstrated in the main text, the empirical covariance matrix of many real world data-sets can be faithfully modelled by a Wishart matrix with long range correlations, where the bulk of eigenvalues is described by the population covariance $\Sigma_{\text{pop}} = \Gamma(1 + \alpha)(i/d)^{-1-\alpha}\delta_{ij}$. As we discussed in App. B,

we can utilize our RMT observations to give a closed formula expression for the empirical Gram matrix eigenvalues and spectral density, in terms of a generalized MP law.

Following this choice of data modelling, and focusing on the bulk eigenvalues alone, it is clear that the sums Eqs. (38) and (39) are the empirical averages over the function $e^{-4\eta_0 \nu t} f(\nu)$, if $\nu$ follows the spectral density derived in App. B. We can the solve the training dynamics by approximating the sum by its respective expectation value,

$$\mathcal{L}_{\mathrm{tr}}(\eta_0, \lambda, \alpha, t) \approx \mathbb{E}_{\nu \sim \rho_{\Sigma_{\mathrm{pop}}}(\lambda, \alpha)} \left[ \nu e^{-4\eta_0 \nu t} \right]. \tag{40}$$

In order to proceed further for the generalization loss, we note that the rotation matrices which form the basis for $\Sigma_{\mathrm{tr}}$ are random unitary matrices, drawn from the Haar measure. This implies that we can glean further information by averaging over training realizations, which will not change the training trajectory at all, but will provide with an average generalization loss $\langle \mathcal{L} \rangle_U$. We utilize the following property of ensemble averaging over unitary random matrices (Nielsen, 2002)

$$\Phi(X) \equiv \mathbb{E}_U[UXU^\dagger] \equiv \int_{\mathcal{U}} d\mu(U) UXU^\dagger = \frac{1}{d_{\mathrm{in}}} \mathrm{Tr}(X)\boldsymbol{I}, \tag{41}$$

where $d\mu(U)$ is the Haar measure. Since the eigenvalue distribution does not change upon this averaging, the average generalization loss can be expressed as

$$\langle \mathcal{L}_{\mathrm{gen}} \rangle_U \approx \mathrm{Tr}(\Sigma_{\mathrm{pop}}) \times \frac{1}{d_{\mathrm{in}}} \sum_i e^{-4\eta_0 \nu_i t}, \tag{42}$$

which can be approximated by its expectation value

$$\langle \mathcal{L}_{\mathrm{gen}}(\eta_0, \lambda, \alpha, t) \rangle_U \approx \mathrm{Tr}(\Sigma_{\mathrm{pop}}) \mathbb{E}_{\nu \sim \rho_{\Sigma_{\mathrm{pop}}}(\lambda, \alpha)} \left[ e^{-4\eta_0 \nu t} \right], \tag{43}$$

completing the dynamical analysis of the loss curves for the model at hand.

Above we gave a toy example of how one may use our results to obtain justified theoretical predictions. Namely, we solved a simple teacher-student model with power law correlated data, and showed that the training dynamics and convergence both depend on the spectral density of the Gram matrices studied in the main text. We obtained analytical expressions for the training and generalization losses.

We stress that on their own, these findings do not attempt to fully explain many aspects of neural network dynamics and generalization, which depend on additional factors beyond the bulk spectrum, such as the large outlier eigenvalues, eigenvectors and higher moments.

Analyzing the interaction between these elements and learning dynamics/generalization remains an important open question, as recent works have started to demonstrate how outliers impact early gradient steps and network collapse.

For instance, as shown by (Seddik et al., 2020) and verified by our results, the outliers can also be described by a Gaussian model, but simply not the CGD that we presented in this work, as the largest eigenvalues are expected to describe the most shared features in the entire data, and do not demonstrate the local correlation structure of the bulk. They are certainly important in classification tasks, and in particular their effect, as well as the effect of the different class mean values are the most important for linear classifiers, as shown in (Saxe et al., 2019).

Our approach focused more on the regime where one would like to understand improved performance using more and more data, where the largest eigenvalues have long been well captured, and the only performance gain that can be achieved is squeezed out of the bulk alone. This has proven a sufficient path to construct solvable models which approximate real-world generalization curves (Kaplan et al., 2020; Maloney et al., 2022; Mei and Montanari, 2022).

