# OpenReview forum: "The Underlying Universal Statistical Structure of Natural Datasets"
_ICML.cc/2025/Conference — ICML 2025 poster_

### Official Review · Reviewer_ZEdS · 2025-03-13

**Overall Recommendation:** 3

**Summary:**

A model of natural dataset (at least natural images) is build based on a random matrix analysis and extensive data analysis. An interpretation is proposed for the power law spectrum of the covariance matrix of input features which generically shows up on natural data. In addition a further characterization of the statistical ensemble is provided in terms of a random matrix ensemble (Gaussian orthogonal ensemble (GOE)) based on a emprirical study of the statistics of level spacing.

**Claims And Evidence:**

The main claims is that the bulk spectrum statistics of the feature-feature covariance matrix  arises from long range correlations following a power law and is in GOE universality class. This claim is backed on a precise data analysis comparing the RMT model obtained with population matrices given by a full band Toeplitz matrix corresponding to a 1d model of homogeneous interactions growing with distance raised to some exponent.

**Essential References Not Discussed:**

No essential reference is missing in my opinion.

**Experimental Designs Or Analyses:**

As discussed already in  the Methods and evaluation criteria section, the experimental data analysis is perfectly sound.

**Methods And Evaluation Criteria:**

On one side the RMT model for the considered 1d model of long range interactions is solved in order to get the corresponding spectral density in the proportional scaling regime (fixed ratio N_data/N_features ) using Marchenko Pastur  equations. It is then used to fit very well the spectral density of a large selection of datasets (image datasets). Then the level-spacing statistics corresponding to these datasets is determined empirically using methods from quantum chaos, namely the unfolding method used to normalize the level spacing w.r.t the spectral density, the distribution of level-spacing statistical indicator called the r-statistics and the spectral two-point  form factor,
which have well defined signature for different RM ensembles. These are then compared with the expected result for GOE yielding a very good agreement. An estimation of the number of samples needed to be in this RMT regime is also provided based on various indicators having well defined values in the matrix ensemble once the exponent is known. I don't see how this can be used in practice, since the exponent which is in principle unknown, is estimated itself at finite M. This looks more like an qualitative heuristic  estimation.

**Other Comments Or Suggestions:**

equation (9) to check, should it not be continuous at tau=1?

**Other Strengths And Weaknesses:**

- strength: the statements on the statistical properties of natural data are interesting and instructive, I found the paper clear and the methodology rather convincing.
- weakness: the scope is limited to natural image and their texture properties. This might have limited impact in ML theory (see my questions below).

**Questions For Authors:**

I find the paper quite interesting but I wonder about the real scope of this work regarding ML in particular the modelling of data for theoretical analysis. Among the claims, the fact that the real datasets fall into the GOE is well expected (Wishart matrices obtained from large data embedded in large dimensional spaces), but the empirical analysis of the level-spacing has the merit of providing a clean validation of this hypothesis. Still, the impact on the theory of ML is not clear to me, since the power-law hypothesis (with possibly some additional spike model of outliers) of the population matrix is already commonly used, while the level-spacing statistics of the Gram matrices  seems not to play (to the best of my knowledge) any role for theoretical analysis of ML. Then remains the 1 dimensional long-range correlated model of the population matrix, in the form of Toeplitz matrices, which looks quite artificial to me. I guess any d-dimensional  model with spatial power-law decay would lead to the same result, but the the 2-point function would decay with distance for d>1 instead of increasing in 1d. I believe this bulk modelling  is a simplified model of texture in this context of image dataset and wonder whether a 2d model would not be more appropriate to interpret the data?

**Relation To Broader Scientific Literature:**

I think the bibliography is fine.

**Theoretical Claims:**

No specific theoretical claims

---

> ### Author Rebuttal · Authors · 2025-04-01
>
> Dear Reviewer ZEdS,
>
> Thank you for your careful review and positive appraisal of our work, tending towards acceptance. We’re glad you found the work interesting and well supported by empirical evidence.
> Below, we address the remaining issues you raised, as well as questions and comments.
>
> **Methods and Evaluation**
>
> *” I don't see how this can be used in practice, since the exponent which is in principle unknown, is estimated itself at finite M. This looks more like an qualitative heuristic estimation.”* - We agree that defining the asymptotic exponent itself at finite $M$ is suboptimal, and therefore its value may not be entirely trustable, but this is truly the best one can do on any real-world finite dataset. The important part is the universal convergence towards this exponent, and the interesting fact that this convergence occurs in two sharply separated phases, as seen in Fig 5 (middle).
>
> **Weaknesses and Questions**
> - *”…level-spacing statistics of the Gram matrices seems not to play (to the best of my knowledge) any role for theoretical analysis of ML.”* - This is inaccurate, please see Appendix F for more details. Briefly, the role of the level spacing metric is to confirm the spectral density distribution, which is critical for analysis of even linear regression in the empirical limit (the resolvent is required to find the optimal network solution, which depends directly on the data covariance matrix’s eigenvalue distribution).
> - *”…whether a 2d model would not be more appropriate to interpret the data?”* - In order to capture the eigenvalue statistics a 1-d model is sufficient, and any higher dimensional model can work, provided that there is a component that decays as a power law. For instance if we were to take a fourier type model, as long as the spectrum had a component that decayed with the magnitude $|\vec{k}|$ as a power law, the same type of results would hold. Our goal was simply to provide the simplest possible model.
> -	“*equation (9) to check, should it not be continuous at tau=1?*” Yes it should, thank you for bringing this to our attention, we will correct the formula in the revised version.
>
> We hope that our replies are sufficient to raise your confidence in our work and possibly raise your scoring of our paper. We welcome any further questions and comments.

---

### Official Review · Reviewer_kWZr · 2025-03-13

**Overall Recommendation:** 4

**Summary:**

The paper examines the empirical eigenvalues of Gram matrices of natural image datasets, showing that they follow a power law. It offers a simple model that reproduces this power law behavior using Gaussian data with long-range correlations. These results suggest that natural image Gram matrices can be approximated by Wishart random matrices with a simple covariance structure, enabling rigorous analysis of neural network behavior.

**Claims And Evidence:**

As far as I can see, all claims are clearly stated and well-supported by theoretical or empirical evidence.

**Essential References Not Discussed:**

• Ba et al. 2022 - High-dimensional asymptotics of feature learning: how one gradient step improves the representation.

• Wang et al. 2023 - Spectral evolution and invariance in linear-width neural networks.

**Experimental Designs Or Analyses:**

I found the experiments to be generally sound and valid.

**Methods And Evaluation Criteria:**

The proposed methods and evaluation criteria make sense for the problem or application at hand.

**Other Comments Or Suggestions:**

Typos and other small issues:

1. Line 153 right column, above eq 2: 'with' is misplaced.

2. I find the index notation confusing, e.g. $X_{ia}\in\mathbb{R}^{d\times M}$; usually $X_{ia}\in\mathbb{R}$ as in the matrix element.

**Other Strengths And Weaknesses:**

Strengths:

1. The paper is well structured, clearly written, and rather easy to follow, given the heavy math.

2. Tackles a very timely subject, both interesting and relevant.

Weaknesses:

1. I think that the main weakness is establishing the claim in the last sentence of the abstract: “image Gram matrices can be approximated by Wishart random matrices with simple covariance structure, enabling rigorous analysis of neural network behavior.” The paper could be much improved if this point was made more central, beyond what is provided in App. F, e.g by some experiments of training a DNN on real-world data vs on the corresponding CGD.

**Questions For Authors:**

1. Lines 434-436: “accomplishing learning tasks for which the spectrum is insufficient and eigenvector properties are needed, requires a different scale of data samples.” - would it not be fair to say that for most typical learning tasks eigenvector properties are needed?

2. What information is NOT captured by the CGD?

3. Can you speculate why the scaling law behaior begins around i=10

4. Lines 189-191: “For real-world data, we consistently find that \alpha>0” - is there a simple reason for this?

**Relation To Broader Scientific Literature:**

I found the subsection “Neural scaling laws” under Background and Related Work somewhat non-satisfactory. It leaves an impression that the relation between the spectral properties of the data and the learning curve is unclear.

**Theoretical Claims:**

I checked the theoretical claims and found no major issues with their correctness.

---

> ### Author Rebuttal · Authors · 2025-03-31
>
> Dear Reviewer kWZr,
>
> Thank you for your careful review and positive appraisal of our work, deeming it acceptable at ICML. We’re glad you found the work interesting and well supported by empirical evidence.
> Below, we address the remaining issues you raised, as well as questions and comments.
>
> **Related work**
>
> 1)	We apologize for not bringing the connection between neural scaling laws, learning curves and the spectral properties of the data more to the forefront in this section. We provide a concrete example for this connection in Appendix F, but we will clarify these connections in the related work section as well in the revised version.
> 2)	Thank you for pointing out these references, we will update the revised manuscript to include them both.
>
> **Weaknesses**
>
> 1)	*Highlighting the connection to real world DNN training on CGD* - We understand the reviewer’s suggestion and agree it has considerable merit, however, our intent in this paper was to bridge a long line of existing semi-empirical works (including ones that train networks on gaussian versions of real world data, see Ref [1,2] and similar works), and purely theoretical works that make the assumption of gaussian data. We will include Ref [1,2] and several other works that perform such experiments to better connect our results to the already large existing literature.
>
> **Comments**
>
> 1.	*Line 153 right column, above eq 2: 'with' is misplaced.* - Thank you for this comment, the typo will be fixed.
>
> 2.	*I find the index notation confusing, e.g. $X_{ia}\in \mathbb{R}^{d×M}$; usually $X_{ia}\in \mathbb{R}$ as in the matrix element.* - We accept your comment, and will augment the final version accordingly.
>
>
> **Questions**
>
> 1.	*Lines 434-436: “accomplishing learning tasks for which the spectrum is insufficient and eigenvector properties are needed, requires a different scale of data samples.” - would it not be fair to say that for most typical learning tasks eigenvector properties are needed?*  - We agree that there are many cases, such as classification tasks, the eigenvectors are crucial, and we study to what extent in a future work. However, for teacher-student settings, many other regression based settings, the eigenvectors are not necessarily important, and the spectrum is sufficient to predict performance and dynamics.
>
> 2) & 3.  *What information is NOT captured by the CGD? why does the scaling law behavior begins around i=10?* - First, the CGD does
>              not capture the top ~10 eigenvalues, since these eigenvalues contain the least “universal” properties of the data, and constitute the most particular aspects of every different dataset, also corresponding to the most informative eigenvectors. Therefore, the CGD, which describes the universal part of the spectrum, is shared between datasets and is only valid for the bulk of eigenvalues. Second, as stated in the main text, the CGD does not capture the eigenvector structure, as we do not assume a special basis for the CGD model. It would be interesting to explore which combinations of bases and spectra would render the CGD model a more complete description of particular datasets.
>
> 4.	*Lines 189-191: “For real-world data, we consistently find that $\alpha>0$” - is there a simple reason for this?* - This is a very good question, which is currently still open. One possibility is that these values of power law spectra can emerge from hierarchical data structures with varying levels of rare features [3]. Answering this question could also shed light on certain aspects of neuroscience, see [4] for the emergence of power laws in human visual cortex activity data. We would also like to note that $\alpha>0$ seems to be prevalent for natural datasets, but not necessarily for other types of data. For instance, simulations of physical systems can give rise to negative exponents (see for instance [5] for the case of turbulence).
>
> References:
>
> [1]  Refinetti et al., Neural networks trained with SGD learn distributions of increasing complexity (2023)
>
> [2] Székely et al., Learning from higher-order correlations, efficiently: hypothesis tests, random features, and neural networks (2024)
>
> [3] Cagnetta et al., How Deep Neural Networks Learn Compositional Data: The Random Hierarchy Model (2024)
>
> [4] Gauthaman et al., Universal scale-free representations in human visual cortex (2024)
>
> [5] Levi et al., The Universal Statistical Structure and Scaling Laws of Chaos and Turbulence (2023)

---

### Official Review · Reviewer_FARz · 2025-03-14

**Overall Recommendation:** 2

**Summary:**

This paper aims to exploit Gaussian universality to generate synthetic data that accurately captures the statistics of natural data distributions. In particular, the paper proposes a synthetic model that reproduces the eigenvalue statistics of the covariance matrices of natural data. Since the synthetic data is described by RMT, the authors focus on demonstrating that the RMT predictions match natural datasets with increasing accuracy as the sample covariance approaches the population covariance.

**Claims And Evidence:**

Claim 1: Powerlaw spectrum of population covariance matrices. This claim is well supported by the experiments and is in agreement with much previous work.

Claim 2: The powerlaw spectrum originates from underlying correlational structure. This claim may be true but it is not supported by the given evidence. See Theoretical Claims section.

Claim 3: The correlated Gaussian datasets are a correct proxy for natural data. This claim is well supported by the experiments, but it follows from the Gaussian universality principle, which has been extensively studied in previous work. See Essential References Not Discussed section.

Claim 4: Convergence of sample covariance to population covariance corresponds to convergence of sample statistics to RMT statistics. I believe this claim already follows from Gaussian universality and law of large numbers.

Claim 5: Shannon entropy is correlated with the local RMT structure, is smaller in correlated datasets, and converges to the population entropy in much fewer samples. The meaning of this claim is unclear. The proposed definition of Shannon entropy does not appear to match the commonly accepted definition for Gaussian distributions. See Methods And Evaluation Criteria section.

To me, the main claim of this paper seems to be that Gaussian universality holds. This has been confirmed in many previous works, so it's not clear what the novel contribution is.

**Essential References Not Discussed:**

The following papers give convincing evidence for Gaussian universality in natural datasets. Many of these concern kernel ridge regression, where the random matrix in question is the kernel matrix \phi(X)^T \phi(X) rather than the covariance XX^T, but the result is analogous (since the spectra of covariance and gram matrices are the same).

https://arxiv.org/pdf/2302.08923
https://arxiv.org/pdf/2102.08127
https://arxiv.org/pdf/2203.06176
https://arxiv.org/pdf/1905.10843
https://arxiv.org/pdf/2311.14646

**Experimental Designs Or Analyses:**

It is known to be notoriously difficult to directly measure powerlaw exponents (see https://arxiv.org/pdf/cond-mat/0402322). Appendices B and C in https://arxiv.org/pdf/2311.14646  also point out that it is easy to make drastic measurement errors trying to directly fit the sample covariance eigenvalues. Due to the ease of measurement errors, I do not know whether to trust the measurements in this paper. The previously mentioned paper provides a more reliable procedure, based in RMT, for exponent estimation.

**Methods And Evaluation Criteria:**

The definition of dataset entropy is confusing to me. I don't see the connection between the entropy as defined and the entropy of the data distribution, which seems the more natural measure of entropy. What motivates the proposed definition?

**Other Comments Or Suggestions:**

Eq 1: I believe it should be X_{ja} or X^T_{aj} instead of X_{aj}

**Other Strengths And Weaknesses:**

My primary concern with this paper is that the main claim seems to be that natural data matrices behave like Gaussian random matrix ensembles when the data dimension is large enough. This is already a well-established claim; there is a broad literature on Gaussian universality in high-dimensional statistical inference problems (see Essential References Not Discussed). This paper suggests some physics-inspired diagnostics for further verifying this claim and find agreement; this is useful but, on its own, I don't think it is a sufficient contribution for this venue.

The strengths include proposing to port over some empirical measures of RMT behavior from systems exhibiting quantum chaos. Another novel idea was to suggest a data generation process via Toeplitz matrices, which produces the anisotropic Gaussian features, but I didn't find it convincing.

**Questions For Authors:**

1. How does your analysis and measurements extend the empirically known Gaussian universality principle?
2. Do the specific measurements you proposed shine light on any specific training phenomena? E.g., learning problems like linear regression or neural networks?

**Relation To Broader Scientific Literature:**

Line 78 "and many advances have been made by appealing to the RMT framework." Give examples?

See Other Strengths And Weaknesses section.

**Theoretical Claims:**

I don't see novel proofs or derivations in this paper. There are some conceptual issues I will point out here.

It's not clear to me why the setup is called "correlated." The data vectors are drawn from a Gaussian with diagonal covariance. It seems that the uncorrelated/correlated distinction being drawn is in fact a distinction between isotropic/anisotropic. Relatedly, it's not clear to me what the Toeplitz matrix brings to the analysis. The exponent in the Toeplitz matrix must be measured from the dataset to be modeled anyways; why not simply use that measurement to directly set the diagonal of the covariance according to eq. 3? With this simplification, I believe the main claim boils down to the usual Gaussian universality assumption.

---

> ### Author Rebuttal · Authors · 2025-04-01
>
> Dear Reviewer FARz,
>
> We appreciate your thoughtful reading of our manuscript. Your main concern is the statement that natural data matrices behaving as Gaussian at sufficiently large dimensions is already known in the literature. We argue that this is **not the case** and we will attempt to explain our estimation of the distinction between what was known and our work below.
>
> **Weakness and Questions**
>
> We believe there is a subtle misunderstanding between us and the reviewer, which may come from the focus of our work. The large body of literature which concerns Gaussian universality concerns the **learning curves of neural networks when taking gaussian data as a surrogate for real data**, or the **equivalence between neural networks in certain limits and Linear noisy networks** applied to **Gaussian data** [1]. None of the works cited by the reviewer deal directly with the statistical structure of the **data itself**. This is a key difference in our opinion. We agree that the studies of generalized linear regression rely on the properties of the covariance matrix which we study, but they are an indirect measurement. Our work then serves two purposes: (1) it is an agnostic, and much stronger validation of the conjectures used in the literature to model the data (2) the r statistics (as well as the other local statistical measures) can offer a systematic path towards explaining the discrepancies between predictions for gaussian and real world data, since $r^{(n)}$ statistics explicitly measure deviations from Gaussianity (3) the local statistics inform you of the number of samples required to reach the RMT regime, showing a non-trivial universal behavior which is not discussed in the previous works (4) the fact that we find level repulsion could have implications for the localization of eigenvectors, which certainly does have an effect for learning tasks, particularly classification tasks. Note that some of the works you cited are already discussed in our text, particularly in App. F, but we should highlight these in the main text.
>
> **Regarding specific questions:**
>
> 1.	*”How does your analysis and measurements extend the empirically known Gaussian universality principle?*” – as we replied in the above paragraph, please see our points (1,2) and (3).
> 2.	*”Do the specific measurements you proposed shine light on any specific training phenomena? E.g., learning problems like linear regression or neural networks?”*  - We have not studied the direct effects of our results on models beyond App. F, but it is clear that this is the key question that will be studied in future work. The intuition is that by studying the higher level spacing statistics on real data we can systematically describe the deviations from Gaussianity. By focusing on the effects of these deviations for learning curves, we should be able to make better predictions for learning curves as described in (2) above. Additionally, see point (4) in the previous paragraph.
>
> **Methods**
>
> The reason for this definition of entropy is a common practice in quantum chaotic systems – we wish to interpret the data matrix as the equivalent of a physical hamiltonian whose complexity can be analyzed using its spectrum. Then, the Renyi entropy definition corresponds to a version of entanglement entropy, measuring the structural complexity of the data.
>
> **Theoretical claims**
>
> We use the “correlated” term simply due to the Toeplitz matrix assumption. The reason for this model is that it gives a simple “real-space” interpretation of the known spectrum. In that sense the source for anisotropy comes from correlations, offering some direction towards understanding why the power law appears at all.
>
> **Experimental design**
>
> We thank the reviewer for bringing this work to our attention. We will incorporate the proposed methods into our work. Still, we would argue that since the errors reported in Simon et al. with direct methods are at most ~5%, our results can be trusted, since we do not focus on the precise estimation of $\alpha$, but rather showing that it is generically between 0 and 1 for natural data. We believe our observations regarding convergence trends and spacings are also reasonable within this  5% error margin.
>
> **Relation to broader literature**
>
> *”Line 78 "and many advances have been made by appealing to the RMT framework." Give examples?”* - We relegated some of the relevant citations to the appendices due to space limitations, we will revise our text to better reflect the contributions in the main text.
>
> **Comments**
>
> Thank you for the comment regarding Eq.1, it will be fixed.
>
> We hope that our replies are sufficient to re-evaluate the merit of our work and change your score of our paper. We welcome any further questions and comments

---

> > ### Comment · Reviewer_FARz · 2025-04-03
> >
> > I'm afraid I still don't understand. The previous works on Gaussian universality and ridge regression (e.g. the one I mentioned earlier, https://arxiv.org/pdf/2311.14646) make claims akin to "some learning algorithms trained on real data behave as if the data were Gaussian with powerlaw covariances." Your work says "real data behave as if they are Gaussian with powerlaw covariances." (E.g., 324L "we show that correlated Gaussian datasets capture [...] the spectral density and the distribution of eigenvalues for ImageNet, CIFAR10, and FMNIST.") In this sense, your results appear to be strictly weaker than what was known before. If there is some subtlety I'm missing, please let me know.
> >
> > Points (2) and (4) would be much more convincing if this analysis (or a preliminary version of it) appeared in the manuscript. (3) is interesting to me, but I don't feel that it is a sufficient contribution for acceptance.
> >
> > "We wish to interpret the data matrix as the equivalent of a physical hamiltonian whose complexity can be analyzed using its spectrum." Why? I'm not understanding what one learns about the data by defining and measuring this quantity. I'm sure one can write down a long list of functionals of the spectrum which converge to RMT predictions w.r.t. dataset size, and it's not clear what's special about this one. I think it adds to confusion, since it contradicts the established information-theoretic definition of entropy that is common in ML.
> >
> > "In that sense the source for anisotropy comes from correlations, offering some direction towards understanding why the power law appears at all." I don't see how. It looks like (reading Eq. 4) the data covariance powerlaw is inherited directly from the Toeplitz powerlaw. From a modeling standpoint, it appears that the questions of "why is the data covariance powerlaw" has not been resolved, only been pushed back to the question "why are the Toeplitz correlations powerlaw." In the absence of more evidence that the Toeplitz picture is the right one, this feels like adding extra complications without any new insight. I'm also uncomfortable with the fact that the exponent for natural data is positive -- if the Toeplitz picture is correct, this implies that the correlations increase without bound, which seems unphysical.
> >
> > **EDIT: In response to reply below.**
> >
> > Thank you for explaining the interpretation of the Gaussian universality claim. I see what you are saying, though I argue that in the end what we (or I) care about is the behavior of some algorithm. Your point is taken, though.
> >
> > Regarding entropy, I agree that the measure you are referring to is interesting for other reasons. Perhaps it would be salient to connect your notion to established notions of effective rank (https://www.eurasip.org/Proceedings/Eusipco/Eusipco2007/Papers/a5p-h05.pdf). (In fact, this paper refers to your quantity as "spectral entropy," which seems a far more apt term than "distribution entropy," which carries a distinct connotation.)
> >
> > I'm still not convinced by the argument regarding the Toeplitz matrices. My initial qualm was with the term "correlated Gaussian data" which seems a misnomer since mean-zero Gaussian data is definitionally uncorrelated. I acknowledge that you derive the spectrum from a correlation argument, but my point is that one can arrive at this spectrum in a variety of other ways that are all in the same "Occam's razor equivalence class" (I made up that term but hopefully it gets the point across). In fact, it seems to me that simply positing that the spectrum is powerlaw is in a *lower* Occam's razor class (it makes the same predictions with fewer steps). This "posit powerlaw spectrum" approach is taken in other works on Gaussian universality. (This is what I meant in my original review by "The exponent in the Toeplitz matrix must be measured from the dataset to be modeled anyways; why not simply use that measurement to directly set the diagonal of the covariance according to eq. 3?") I didn't know about the increasing correlation of velocity differences within the eddy-size spatial scale, thanks for pointing that out.
> >
> > I've increased my score to 2 due to these discussions.

---

> > > ### Author Response · Authors · 2025-04-07
> > >
> > > Thank you for your continued engagement and willingness to discuss your concerns. We hope the following addresses your concerns:
> > >
> > > The first statements you made accurately describe the observations, but your conclusion from these observations is the opposite of ours. Perhaps a simple way to see why we claim our statement is stronger than just looking at the learning curves is the following observable counting argument: the loss/learning curves constitute an averaged, scalar observable over the data (in the case of ridge regression). This averaging, as well as the fact that it is a single number, makes it quite insensitive. Concretely, any deviation of the data from GOE  will manifest as a small correction to a single number (the loss). On the other hand, measuring the level spacing distribution, r-statistics, SFF etc., constitute a much larger number of **direct** measurements of the data. All of the local statistics distributions have moments, and all of them can computed and compared with the data. By constraining all of these observables we offer a new, much more quantitatively controlled approach to understanding natural datasets, agnostic of any neural network training. Additionally, we can distinguish between GOE and other closely related universality classes such as the short range plasma model [1], which could fall below the sensitivity of a single scalar observable (learning curves). We sincerely hope that this clarifies why we believe directly measuring the data is a much stronger statement than any implicit measurement (such as measuring the loss of networks).
> > >
> > > **Further analysis:** Thank you for acknowledging that points (2) and (4) are interesting. We have done a preliminary analysis of the $r^{(n)}$ statistics for $n=1,2,4,6$, for CIFAR 10 and for samples from a GOE data, which can be viewed in this link: https://imgur.com/a/PGb7Dfh.
> > > We see that the nearest neighbor statistics agree very well with the GOE, but as the value of $n$ gets larger, we probe farther neighbors and we start seeing discrepancies. We will perform a more complete analysis and add it to the appendix, highlighting this as a possible new avenue to study GOE in datasets. Regarding point (4), we refer the reviewer to App. E, where we have begun an analysis of the eigenvector/localization structure. We have future work along this direction, but it seemed to us that it warrants its own work rather than being a part of this one.
> > >
> > > **Regarding entropy:** The eigenvalues of a matrix tell you how its action is distributed across different orthogonal directions, the eigenvectors. An entropy built from those eigenvalues quantifies whether the matrix is dominated by one or a few large eigenvalues, or whether its eigenvalues are more evenly spread out. We don’t mean to conflate this with the standard information theory definition of entropy, and will clarify this in the text.
> > >
> > > **Regarding our interpretation of the Toeplitz matrix correlation structure:** We do not aim to interpret the correlation structure in the 2 dimensional space of the pixels, since obviously these correlations should not be growing ones. Instead, these are correlations (that do grow) in a 1 dimensional space that should somehow be related to the features of the data. Understanding why this particular structure forms is indeed a different way of asking the question of why power laws are generated, but we think additional ways of phrasing the question could lead to interest results. Additionally, we note that non-unitary theories such as turbulence, where the Kolmogorov scaling exponents show that correlations of velocity differences grow with the separation distance in the inertial range of scales have negative $\alpha$, while real world datasets have positive $\alpha$. That suggests that we should interpret differently the correlation, and that real data correspond to classes of unitary physical models.
> > >
> > > Thank you again for the discussion and useful insights. We would gladly have continued discussing if not for the constraints of the review process. We hope that we have answered your questions, and if so, we would greatly appreciate raising our score towards acceptance.
> > >
> > > References:
> > >
> > > [1] Rao, W., Critical Level Statistics at the Many-Body Localization Transition Region (2021)

---

### Official Review · Reviewer_LgA8 · 2025-03-25

**Overall Recommendation:** 3

**Summary:**

The paper investigates the universal statistical structure underlying the covariance matrices (Gram matrices) of natural datasets. Leveraging Random Matrix Theory (RMT), it demonstrates that real-world datasets and correlated Gaussian datasets (CGDs) share universal spectral properties, including a characteristic power-law decay of eigenvalues and chaotic statistical behavior consistent with Gaussian Orthogonal Ensemble (GOE) universality. This universal behavior suggests that natural data covariance structures can be effectively modeled by simpler correlated Wishart matrices, enabling analytical tractability and better understanding of neural network learning dynamics.

**Claims And Evidence:**

Extensive numerical experiments across multiple datasets, rigorous statistical tests, and clear comparisons between empirical data and theoretical models from RMT support the claims.

**Essential References Not Discussed:**

Key related works essential to understanding the paper include studies on neural scaling laws (Kaplan et al., 2020; Maloney et al., 2022), foundational papers in RMT (Mehta, 1991; Couillet and Liao, 2022), and previous analyses of data spectral structure (Ruderman, 1997; Pennington & Worah, 2017).

**Experimental Designs Or Analyses:**

Experiments are thorough, involving various real-world datasets (MNIST, FMNIST, CIFAR-10, ImageNet), uncorrelated Gaussian data, and correlated Gaussian datasets. The methodology is clearly explained and robustly implemented. However, experiments are limited to image domain only and it would be interesting to see how the findings extend beyond image modality.

**Methods And Evaluation Criteria:**

NA

**Other Comments Or Suggestions:**

NA

**Other Strengths And Weaknesses:**

Clear identification of a universal statistical structure in real-world datasets.

Strong theoretical underpinning via RMT.

Rigorous and comprehensive experiments.

**Questions For Authors:**

How robust is the observed universality across modalities other than vision (e.g., natural language or audio)?

Could different neural architectures affect the emergence of universal spectral structures?

What are the practical implications or potential limitations when applying these universal properties to realistic neural network training scenarios?

**Relation To Broader Scientific Literature:**

The paper effectively situates its findings within the broader context of neural scaling laws, chaos theory, and Random Matrix Theory. It provides a novel connection between empirical observations of natural datasets and theoretical models traditionally used in quantum chaos and statistical physics.

**Theoretical Claims:**

Yes, the theoretical claims appear sound. The authors effectively apply well-established RMT concepts and derive theoretical predictions, which closely match the empirical data.

---

> ### Author Rebuttal · Authors · 2025-03-30
>
> Dear Reviewer LgA8,
> Thank you for your positive reading of our manuscript, deeming our paper acceptable.
>
> Below, we address your comments/questions:
>
> **Essential References Not Discussed:**
>
> We believe the list offered is likely a mistake, since all of these references are explicitly cited in our related work section under “Random Matrix Theory”. Please let us know if we have missed something.
>
> **Questions**
>
> 1)	*How robust is the observed universality across modalities other than vision (e.g., natural language or audio)?* - The results appear to be quite universal, also across different modalities. In particular, we studied the RMT properties of language data, and found similar results. These results will be postponed to future work.
> 2)	*Could different neural architectures affect the emergence of universal spectral structures?* - Since the essence of our work deals directly with the data itself, the notion of a neural architecture doesn’t explicitly enter into our analysis. However, it is clear (and known) that performing different nonlinear transformations on data, can dramatically change its RMT behavior, since it essentially changes the moments of the underlying data distribution, leading to different sample complexities and training dynamics.
> 3)	*What are the practical implications or potential limitations when applying these universal properties to realistic neural network training scenarios?* - The practical implications could range from different sampling techniques based on PCA (and eigenvector sampling) and obtaining scaling laws for sampling required to reach ergodicity, to a more detailed study of the structure of data required for successful learning, moment by moment (second, third etc.).
>
> We hope that our replies are sufficient to raise your confidence in our work and accept the paper. We welcome any further questions and comments.

---

### Decision · Program_Chairs · 2025-05-01

**Decision:**

Accept (poster)

**Comment:**

This paper investigates the universal statistical structure underlying the covariance/Gram matrices of natural datasets, and compares empirical findings against theoretical predictions derived from statistical physics and random matrix theory.

The authors have done a commendable job during the rebuttal phase, and all reviewers are now generally **convinced of the significance and merit of the work**.
It is important to note that this paper does not aim to fully resolve the question of universal structures in natural data.
Rather, it represents a first step toward a deeper understanding of this fundamental and challenging problem.

As highlighted by Reviewer FARz and others, there remains ample room for improvement, including providing novel theoretical analysis and further clarifying the central ideas and contributions.
Nonetheless, given the importance of the problem and the value of this initial contribution, I believe the paper is a meaningful addition to the ICML community.

As a result, I recommend acceptance of this paper to ICML 2025.
Please ensure that the additional numerical results and clarifications provided during the rebuttal phase are incorporated into the final version of the paper.